# Characterization of *Medusavirus* encoded histones reveals nucleosome-like structures and a unique linker histone

Chelsea M. Toner [1,3], Nicole M. Hoitsma [1,2,3], Sashi Weerawarana[1] & Karolin Luger [1,2] ✉

The organization of DNA into nucleosomes is a ubiquitous and ancestral feature that was once thought to be exclusive to the eukaryotic domain of life. Intriguingly, several representatives of the Nucleocytoplasmic Large DNA Viruses (NCLDV) encode histone-like proteins that in Melbournevirus were shown to form nucleosome-like particles. *Medusavirus medusae* (MM), a distantly related giant virus, encodes all four core histone proteins and, unique amongst most giant viruses, a putative acidic protein with two domains resembling eukaryotic linker histone H1. Here, we report the structure of nucleosomes assembled with MM histones and highlight similarities and differences with eukaryotic and Melbournevirus nucleosomes. Our structure provides insight into how variations in histone tail and loop lengths are accommodated within the context of the nucleosome. We show that MM-histones assemble into tri-nucleosome arrays, and that the putative linker histone H1 does not function in chromatin compaction. These findings expand our limited understanding of chromatin organization by virus-encoded histones.

Within the domain of eukaryotes, the compaction of genomic DNA by histones to form nucleosomes is an omnipresent and ancestral feature. The eukaryotic nucleosome core contains four unique histones (H2A, H2B, H3, and H4), each consisting of a structurally conserved histone fold that is common to all four core histones, as well as histone fold extensions (or loops) and highly charged histone tails that are unique to each histone. Two copies each of H2A-H2B and H3–H4 heterodimers assemble into octameric particles that wrap 145–150 bp of DNA to form a canonical nucleosome[1,2]. The histone fold domains of all eight histones are responsible for organizing ~120 bp of DNA, while the N-terminal α-helix of H3 organizes the terminal ~13 bp of DNA on either side[3]. Histone fold extensions further define the surface of the nucleosome. Numerous protein–protein and protein-DNA interactions within the histone octamer produce a stable disc-shaped particle; however, this conformation is more dynamic than originally suggested by the early

crystal structures, and this dynamic behavior is essential for the regulatory function of chromatin[1,4].

The four core histone genes are among the most evolutionarily conserved sequences in the eukaryotic domain of life, suggesting that their incorporation into genomes was an early and essential event during eukaryogenesis[5,6]. There are several hypotheses to explain the enigmatic emergence of the eukaryotic nucleus. Based on the similarities in the information-processing machinery between archaea and eukaryotes, the dominant theory suggests that eukaryotes arose through the metabolic symbiosis of an archaeal host and a proteobacterium, where the gene encoding the single, tail-less histone fold protein of archaea diversified and expanded to give rise to the four core histone genes[5,7–9]. However, the discovery of nucleocytoplasmic large DNA viruses (NCLDV), some of which encode their own histone-like proteins, provided some support to the Viral Eukaryogenesis hypothesis. This hypothesis suggests that the early eukaryotic cell was

[1]Department of Biochemistry, University of Colorado at Boulder, 80309 Boulder, CO, USA. [2]Howard Hughes Medical Institute, Chevy Chase, MD, USA. [3]These authors contributed equally: Chelsea M. Toner, Nicole M. Hoitsma. ✉e-mail: karolin.luger@colorado.edu

a tripartite conglomerate of an archaeon, an alpha-proteobacterium, and a complex DNA virus (possibly represented by modern NCLDV)[10–14].

The diverse life cycles of NCLDV or 'giant viruses' further support this hypothesis by demonstrating various intermediate dependencies on their hosts. They either replicate and assemble in a viral factory that is located in the cytoplasm, or they transiently recruit various host nuclear proteins to the viral factory in the cytoplasm or nucleus for transcription of early genes[15,16]. Additionally, many NCLDV encode homologs of the critical m7G capping apparatus that is absent in most archaea[17]. In all eukaryotes, the nuclear membrane separates chromatin from the cytoplasm and ribosomes, facilitating the decoupling of transcription from translation with m7G capping. These hallmark eukaryotic genes that are present in the virus represent one critical component of eukaryotic differentiation not widely seen in other domains of life[12]. Therefore, while most giant viral proteins were initially believed to have eukaryotic origin, this alternative hypothesis suggests that they may instead have contributed to modern eukaryotic features. As such, investigating histone-encoding NCLDV may provide insight into their role in the debated origin of the eukaryotic nucleus[18].

To date, the ability of virus-encoded histones to form nucleosome-like structures has been demonstrated for only one giant virus (Melbournevirus, a member of the *Marseilleviridae*). The similarity of Melbournevirus and eukaryotic nucleosomes (despite the fusion of viral histones H4 with H3, and H2B with H2A) suggests that histones were either acquired from an eukaryotic host or that eukaryotic nucleosomes (eNuc) have evolved from virally encoded histones[19–21]. Distinct differences not seen in any of the eukaryotic nucleosome structures (e.g. the fusion of H4 with H3, and H2B with H2A, and appropriation of histone tails to modulate nucleosome structure) suggest adaptation of histone sequences to the unique requirements of histones in the Melbournevirus life cycle. As such, elucidating the structure of virally encoded nucleosomes from distantly related giant viruses is of interest to understanding these adaptive mechanisms.

*Medusavirus medusae* (MM), named for its ability to turn the *Acanthamoeba castellanii* (*A. castellanii*) host into stone through encystment, is one of the few NCLDV known to encode all four core histones (H2A, H2B, H3, and H4) on separate genes. Uniquely, the genome also harbors a putative homolog of the linker histone H1[22,23]. Unlike Melbournevirus, the MM genome enters the host nucleus to initiate DNA replication, while particle assembly and DNA packaging are carried out in the cytoplasm near the nuclear membrane[22]. The MM putative linker H1 is expressed with immediate early genes in the MM life cycle, suggesting a role in reshaping host transcription patterns[22,24]. In contrast, the four core MM histones are expressed at a later time-point in the infection cycle, suggesting a role in compacting, protecting, and regulating the viral genome by forming nucleosome-like structures within the established viral factory[22,24]. Compared to eukaryotic and Melbournevirus histones, MM histone sequences have several distinct features, particularly the length of histone tails and loops within the histone fold (Figs. 1a, b, Supplementary Fig. 1A–C). Additionally, the MM putative linker histone H1 has a dramatically more acidic isoelectric point (pI) than most eukaryotic linker histones (Supplementary Fig. 1D–F). Investigation of the MM histones, including the unique linker histone H1, and their ability to organize chromatin expands our limited knowledge of chromatin organization by histone-encoding NCLDV.

We utilized cryogenic electron microscopy (cryo-EM) to reveal that the predicted viral core histones from MM form octamers that assemble with DNA into nucleosome-like particles (MM-NLPs). These NLPs are characterized by unique accommodations for elongated loops and tails, as well as a more pronounced positively charged DNA-interacting ridge compared to eukaryotic histone octamers. Additionally, we demonstrate that MM histones can form positioned tri-

nucleosomes[25]. AlphaFold and atomic force microscopy (AFM) analysis demonstrates that the putative virally encoded linker histone H1 consists of two winged-helix domains with bimodal charge distribution, but does not promote chromatin compaction, suggesting an alternate virus-specific function. As only the second known structure of a viral-encoded nucleosome, this data advances our understanding of how histones have adapted to NCLDV chromatin organization.

## Results

### MM core histones form distinct, stable nucleosome-like particles irrespective of DNA sequence

The MM genome harbors genes for homologs of histones H2A, H2B, H3, and H4 (ORF 318, ORF 61, ORF 255, and ORF 254, respectively)[22]. Secondary structure predictions indicate that MM histones have canonical histone folds ($\alpha 1$–L1–$\alpha 2$–L2–$\alpha 3$, Fig. 1a). Additionally, many signature histone residues are conserved in Medusavirus histones, including arginine side chains that extend into the DNA minor groove, the paired L1 loops (R-T pairs), and intermolecular histone fold stabilization (R-D clamp). Moreover, key eukaryotic nucleosome (eNuc) features, such as the H3 $\alpha$N-helix (which organizes DNA ends of nucleosomes) and the H2B $\alpha$C helix (which defines the surface of the nucleosome), are also present in MM-histones sequences.

The MM histones share a 77.8% sequence identity with histones of its closest relative *Medusavirus stheno*, but <30% with eukaryotic histones, and only 23% with the fused histones from Melbournevirus (the only experimentally determined viral nucleosome structure) or histones from MM's closest relative Clandestinovirus (Fig. 1b and Supplementary Fig. 1). Compared to eukaryotic histones, MM histones $\alpha$-helices are often connected by longer loops (MM-H2B) or have longer tails (MM-H2B, MM-H2A, MM-H3) (Fig. 1a). Specifically of interest, MM-H2A diverges in sequence from eukaryotic H2A in the docking domain, which tethers the H2A-H2B dimers to $(H3–H4)_2$ tetramers, and in the presence of a unique C-terminal extension on histone H3 (Fig. 1a and Supplementary Fig. 1).

We expressed, purified, and refolded the four viral histone homologs into an octameric complex (Fig. 2a). Utilizing the well-established salt-gradient nucleosome reconstitution protocol, increasing amounts of MM octamer were combined with 207 bp DNA (Widom 601 nucleosome positioning sequence) to obtain defined NLPs[26,27]. MM-NLPs migrate much higher on native gels compared to eukaryotic nucleosomes (eNuc), and even higher than Melbournevirus NLPs (Fig. 2b). We confirmed that MM-NLPs on 207 bp DNA (MM-$NLP_{207W}$) contain a full complement of histones by analyzing sucrose gradient fractions by SDS-PAGE, as well as by analytical ultracentrifugation (Fig. 2c, Table 1, and Supplementary Fig. 2).

Sedimentation velocity analytical ultracentrifugation (SV-AUC) allows the determination of macromolecule size and overall shape in solution from diffusion-corrected sedimentation values, which are proportional to particle mass and inversely proportional to viscous drag[28]. Here, eNuc$_{207W}$ sediments at 12 S, consistent with previously published eNuc$_{147W}$ sedimentation at 11 S[19]. MM-NLP$_{207W}$ sediments at ~8 S. This decrease is likely caused by an increased viscous drag compared to eNuc, and is similar to what was observed for Melbournevirus NLP$_{207W}$[19] (Table 1). Reconstituting either MM-(H2A-H2B) dimer or MM-$(H3–H4)_2$ tetramer onto the same DNA gives rise to particles with the expected molecular weight and an even higher level of viscous drag ($f/f_0$). The experimentally determined molecular mass of the largest population (91%) of the MM-NLP$_{207W}$ from SV-AUC (291 kDa) is in close agreement with the expected theoretical mass of an octameric protein complex with 207 bp DNA (304 kDa) (Fig. 2c and Table 1). However, the MM-NLP$_{207W}$ sample displayed a wider range of experimentally determined molecular weights suggestive of a mixture of fully folded nucleosomes and sub-nucleosome species (hexasomes, tetrasomes), reflecting reduced stability of the MM-NLP$_{207W}$ under centrifugal force compared to eNuc (Table 1).

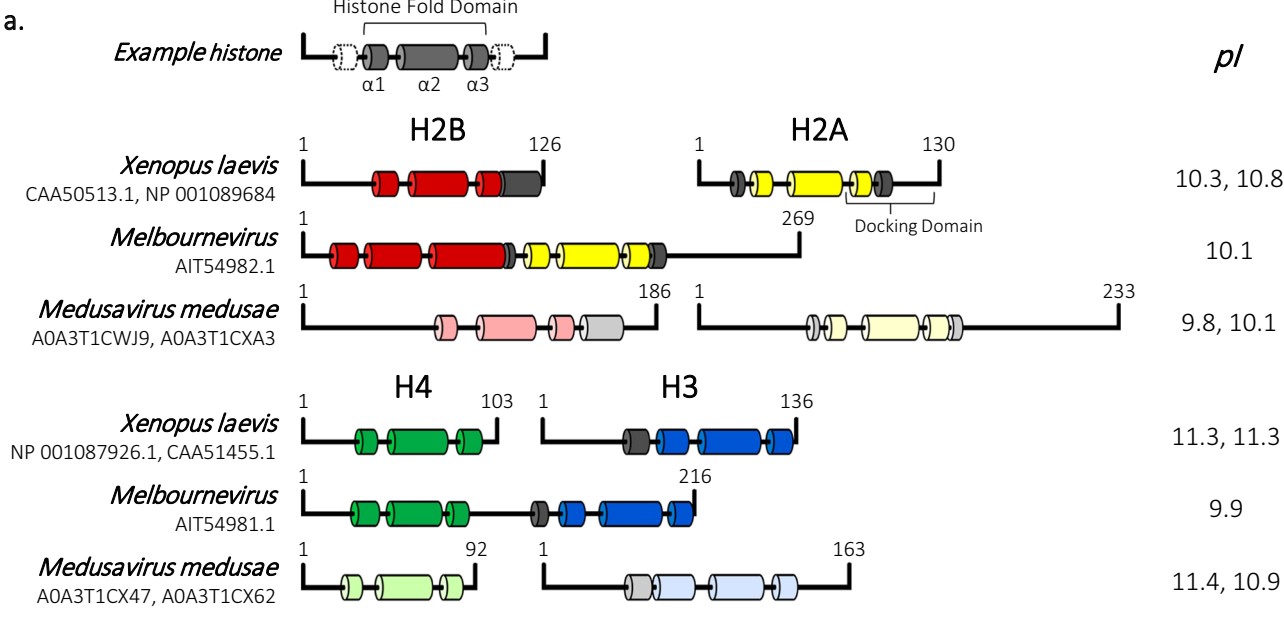

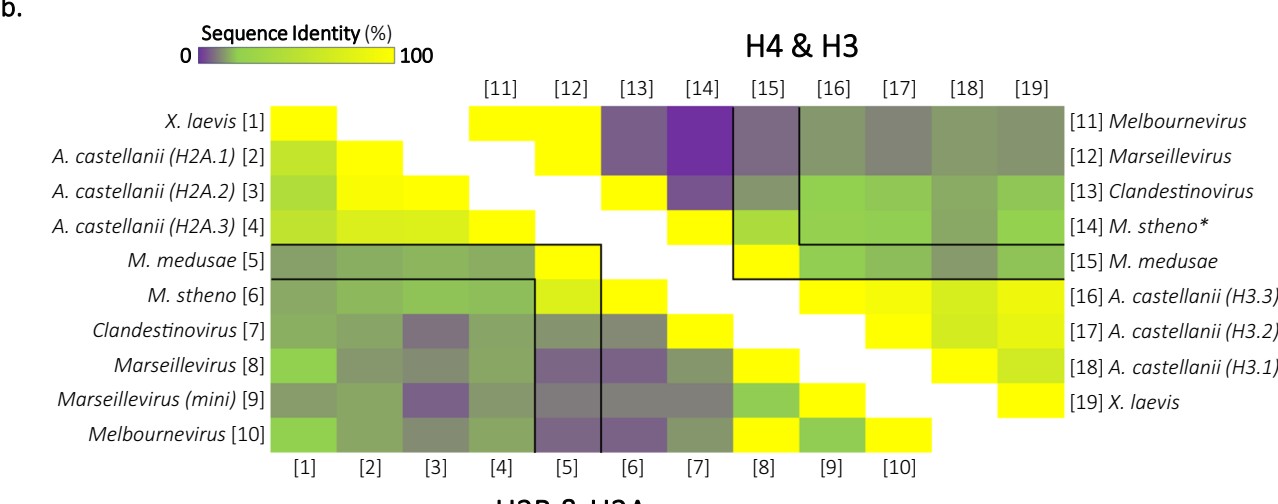

**Fig. 1 | Secondary structure prediction and sequence alignment of *Medusavirus medusae* histones reveals conservation of key eukaryotic residues. a** Schematic of Eukarya (*Xenopus laevis*) and Nucleocytoviricota histones from Melbournevirus (MV) and Medusavirus (MM). Known *X. laevis* and MV α helices representative of the histone fold domain are represented in dark-colored tubes (H2B, red; H2A, yellow; H4, green; H3, blue; and additional helices, gray). α helices in MM histones were predicted using HHpred's Quick 2D prediction webserver (shown in lighter designated colors). Isoelectric points (pI) of each histone are shown to the right. **b** Heat map comparing percent identity of Eukarya and Nucleocytoviricota H2B-

H2A histone sequences (left triangle) to each other. Equivalent H2B-H2A histone sequences are represented along the bottom side [1–10]. Heat map (right triangle) of percent identity of Eukarya and Nucleocytoviricota H4-H3 sequences to each other. Equivalent H4-H3 histone sequences are represented along the top side [11–19]. MM histones are outlined in black within both triangles. A comparison of different dimer pairs (H2B-H2A to H4–H3) sequence identity is not displayed. *\*M. stheno* H4-H3 alignment values determined from H3–H4 alignment shown in Supplementary Fig. 1C. Source data are provided as a Source Data file.

To determine if MM-NLP formation is affected by either DNA sequence or length, we reconstituted MM octamer with a 150 bp 'random' DNA sequence (MM-NLP$_{150R}$). This DNA was designed to have a 50% GC content over each 10 bp segment and does not harbor strong nucleosome positioning signals. Both Medusavirus and eukaryotic histone octamers formed nucleosomes on these sequences. As was observed for the Widom 601 nucleosome positioning sequence, MM-NLP migrate much higher than eNuc on native-PAGE, supporting the finding that MM histones form less compact particles than eNuc irrespective of DNA sequence (Fig. 2d).

The stability of the nucleosomes and their corresponding histone octamers were tested in a thermal melting assay (25 °C–95 °C)

by monitoring the fluorescence of SYPRO Orange release from denatured complex. The MM octamer fluoresces at 25 °C, indicating instability, as opposed to the eukaryotic octamer that does not begin to dissociate until ~40 °C (Supplementary Fig. 3A). eNucs on both DNA fragments (207 W and 150 R) demonstrate the characteristic peaks of histone dimer and tetramer release from the nucleosome, as previously reported[29]. In contrast, all viral NLPs (MM-NLP$_{150R}$, MM-NLP$_{207W}$, and Melbournevirus NLP$_{207W}$) melt in a single peak at much lower temperatures, suggesting the dissociation of the octamer from DNA in a single step, irrespective of sequence and length. This demonstrates the lower thermal stability of both viral octamers and nucleosomes compared to their

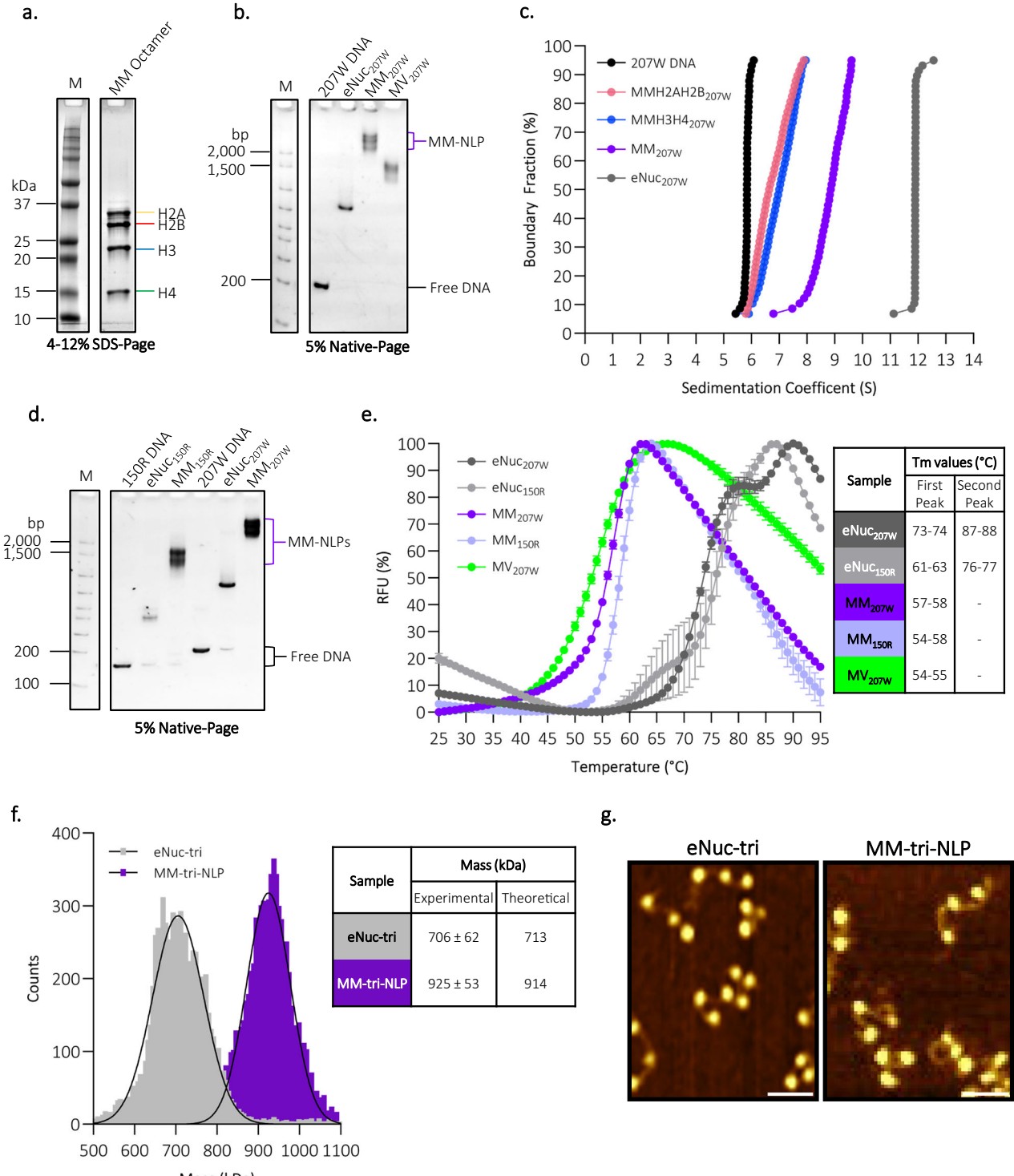

**Fig. 2 | *Medusavirus medusae* histones and DNA assemble into stable mono nucleosome-like particles (NLP) and tri-NLP in vitro. a** SDS-PAGE of refolded *Medusavirus medusae* (MM) core histone octamer (H2A, H2B, H3, and H4). **b** MM-NLP, eNuc, and Melbournevirus NLP (MV-NLP) reconstituted with Widom 601–207 bp DNA and analyzed by 5% Native-PAGE stained with SYBRGold (DNA visualization). **c** SV-AUC of reconstituted NLP and histone-DNA complexes. Van Holde–Weischet plot of eNuc on 207 bp DNA, histone-DNA complexes with H2A-H2B and (H3–H4)$_2$, and MM-NLP$_{207W}$. Quantitative evaluation is given in Table 1. **d** 5% Native-PAGE of reconstituted MM-NLP and eNuc on 150 bp 'random sequence DNA' (50% G/C DNA (150 R)), and on Widom 601–207 bp DNA. **e** Thermal stability of MM-NLP and eNuc shown in **d** including MV$_{207W}$ from **b**. All points are the mean

(SEM as error bars, *n* = 3). Tm values of each MM-NLP and eNuc are shown in the inset. **f** Mass photometry analysis of eNuc-tri (gray) and MM-tri-NLP (purple). The solid lines represent the Gaussian function fit the main species observed on particle counts versus molecular mass distribution histograms, with the estimated molecular weight (in kDa) corresponding to the respective mass at the centre of each peak (i.e., mean ± SD ($\mu \pm \sigma$)). Theoretical and measured molecular masses are shown in the inset. This is one representative data set of triplicate MP measurements. **g** Representative AFM topography images of eNuc-tri and MM-tri-NLP. Scale bar = 50 nm. Unless otherwise noted, all experiments have been repeated independently more than three times with similar results. Source data are provided as a Source Data file.

**Table 1 | S values (s(20, w)), frictional ratios (f/f0), and calculated molecular weights of the subspecies with the largest population (with confidence intervals for the entire sample) of histone-DNA complexes from SV-AUC**

| Sample | s(20,w) | f/f0 | Molecular weight (kDa) experimental | Molecular weight (kDa) theoretical | Protein Complex:DNA |
|---|---|---|---|---|---|
| 207 W | 5.74 (5.70, 5.79) | 3.59 (3.46, 3.77) | 136.84 (122.39, 148.21) | 127.82 | - |
| MM-H2A-H2B$_{207W}$ | 5.81 (5.77, 5.85) | 2.95 (2.86, 3.06) | 195.25 (186.43, 204.06) | 180.22 | 2:1 |
| MM-H3–H4$_{207W}$ | 5.95 (5.94, 5.96) | 2.93 (2.91, 2.95) | 187.89 (185.12, 188.14) | 199.19 | 2:1 |
| MM-NLP$_{207W}$ | 8.21 (6.97, 9.45) | 2.51 (1.85, 3.18) | 290.92 (237.99, 342.84) | 304.00 | 1:1 |
| eNuc$_{207W}$ | 12.10 (12.02, 12.15) | 1.77 (1.69, 1.86) | 253.57 (249.09, 266.22) | 237.28 | 1:1 |

All values were calculated using UltraScan[55,56]. Source data are provided as a Source Data file.

eukaryotic counterparts, as previously shown for Melbournevirus NLP (Fig. 2e)[19].

To test if MM-NLP can assemble into higher-order chromatin, we reconstituted MM tri-nucleosome-like particles (MM-tri-NLP) onto a 3x copy of Widom 207 DNA[30]. As observed for the single MM-NLP, MM-tri-NLPs migrate higher on a native gel relative to *X. laevis* tri-nucleosomes (eNuc-tri) (Supplementary Fig. 3B). The mass of the MM-tri-NLP was estimated in solution via mass photometry to be 925 kDa, highlighting the considerable size difference of the viral histones compared to eNuc-tri at 706 kDa (Fig. 2f). MM-tri-NLPs exhibit the same characteristic "beads on a string" structure on this nucleosome positioning sequence as eukaryotic tri-nucleosomes, forming three distinct NLPs as visualized by AFM (Fig. 2g). Consistent with less stable nucleosomes, the MM-tri-NLP appear more heterogenous than eNuc-tri via AFM analysis, likely due to the formation of nucleosomal subspecies during salt-gradient dialysis or disassociation during dilution (Supplementary Fig. 3C, D).

### MM-NLP make structurally unique accommodations for longer histone tails and loops

Single-particle cryo-EM was utilized to determine the structure of MM-NLP$_{207W}$. Samples were crosslinked through gradient fixation (GraFix) with glutaraldehyde[19], and compared to a native sample that had not been subjected to crosslinking. Migration through the sucrose gradient was unchanged upon crosslinking (Supplementary Fig. 2). Data for both samples was collected on a Titan Krios G3i to obtain electron density maps of crosslinked MM-NLP$_{207W}$ at 3.3 Å (Supplementary Table 1, Fig. 3 and Supplementary Fig. 4A) and native MM-NLP$_{207W}$ at 4.9 Å (Supplementary Fig. 4B). We observed electron density for all core histone helices and ~135 bp of DNA, allowing assignment of the MM-NLP$_{207W}$ core. Density for the MM-H2A tail region of the docking domain was not visible, indicating disorder due to lack of interactions with the remainder of the structure (Supplementary Figs. 4 and 3A).

Just like its eukaryotic counterpart, MM-NLP$_{207W}$ contains two copies each of H2A, H2B, H3, and H4 as an octameric core formed by histone fold regions, wrapped by DNA in a left-handed superhelix (Fig. 3a). The density of DNA termini associated with one of the two H3 αN-helices is very weak, underscoring the dynamic character of the ~13 penultimate base pairs of DNA, also observed in several eukaryotic and Melbournevirus nucleosome structures[31,32]. The overall geometry of the DNA superhelix and the layout of histone fold helices are near-identical between MM-NLPs and the eukaryotic nucleosome (RMSD for histone core, 1.9 Å), with minor differences in the wrapping of DNA ends (Fig. 3b). The structure of native (i.e., not crosslinked) MM-NLP$_{207W}$ shows a high cross-correlation coefficient of 0.87 with crosslinked MM-NLP$_{207W}$, confirming that there were no induced and potentially artificial conformations due to crosslinking (Figs. 3c and 4a).

While the many canonical histone features are conserved between eNuc and MM-NLP (for example, the 'sprocket arginines' that reach into the minor groove of the DNA, Fig. 4b), MM histones have distinctly longer tails (H2A, H2B, H3) and loops (H2B). Most unusually, MM-H3

has a 29 amino acid long C-terminal extension that is not observed in any other H3 histone from any species (Fig. 4b). This region forms a 9 amino acid α helix that extends from the end of α3 and lays across its partnered H4 α2-helix and H3 α1-helix, then extends to reach the DNA minor groove at Super Helical Location (SHL) ± 1.5 (i.e., -15 bp from the central base pair denoted with φ in Figs. 3 and 4). The electron density in this region is well-defined and consistent across both the crosslinked and native structures, indicating that the observed features are not due to crosslinking artifacts (Fig. 4a). This additional helix further stabilizes the H3–H4 heterodimer through a combination of hydrophobic packing and hydrogen bonding interactions. It also redirects the MM-H4 N-terminal tail to contact the DNA at SHL ± 1.5 (Fig. 4b).

A second unique feature of MM histones is the extended H2B L1 loop connecting a shortened α1 and α2 helix of the H2B histone fold. This loop, which is 11 amino acids longer compared to eukaryotic histones, resembles a β hairpin and has a well-defined density that protrudes from the DNA superhelix by -15 Å and extends beyond the equivalent region in eNuc by 7 Å (Figs. 3b and 4c)[33]. The base of this loop packs against H2A by forming a hydrophobic core centered around H2B F91, H2A I126, and H2A I111, and enforced by several potential main chain–side chain hydrogen bonds between H2A and H2B (Fig. 4c). Together with H2B α1, this extended loop forms a defined module that likely contributes to stabilizing the H2A-H2B heterodimer, and uniquely defines the surface of the MM-NLP. Together, the extended H3 C-terminal region and the extended H2B L1 loop (both unique to MV-histones) appear to contribute to stabilizing heterodimer interactions between H3–H4 and H2A-H2B, respectively.

### Medusavirus NLPs share common features with Melbournevirus NLPs, but also have unique elements

To date, the only other viral NLP for which structural information is available is the Melbournevirus nucleosome, where histones are fused into doublets (H2B-H2A and H4-H3)[19,21]. Even with histone doublets, the Melbournevirus NLP structure is quite similar to the eNuc in the positioning of the histone fold elements (RMSD for histone core, 3.7 Å). Therefore, both viral (MM and MV) NLPs are similar in overall shape to the canonical eNuc. An intriguing commonality between the viral NLPs is the presence of well-defined density that lies across the H4 α2-helix (Fig. 5a). In MM-NLP, as described above, the extended H3 C-terminal α helix (so far only found in MM-H3) reaches over H4 α2, while the N-terminal tail of H4 reaches across the same region on the H4 α2-helix in Melbournevirus NLP (centered around MM-H4 L52; Fig. 5a). The amino acids comprising this interface are not conserved between Medusa- and Melbournevirus histones, indicating that this stabilizing interaction was acquired independently in the two viruses, using two different structural elements. In both structures, the interface consists of charge-charge interactions and hydrophobic packing between the H4 α2-helix and the respective tail, indicating an intriguing functional convergence between the two viral particles (Fig. 5a).

The H2A docking domain is a ladle-shaped configuration that tethers H2A-H2B to the (H3–H4)$_2$ tetramer through numerous interactions and, therefore, has a key function in stabilizing the entire

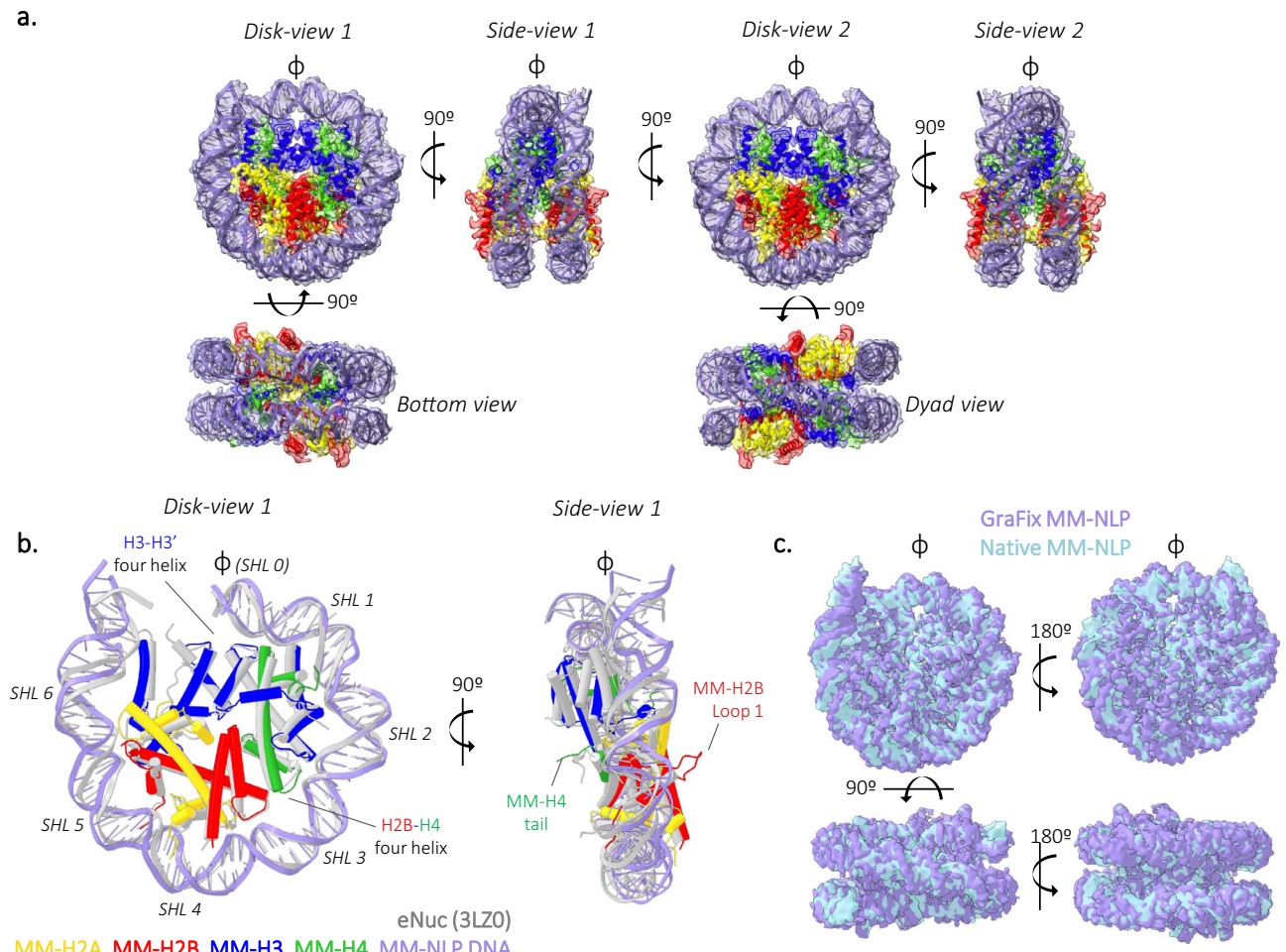

**Fig. 3 | *Medusavirus medusae* NLP (MM-NLP$_{207W}$) resemble eukaryotic nucleosomes. a** Overview of MM-NLP$_{207W}$ structure. Individual histones H2A, H2B, H3, and H4 and their surrounding density are shown in yellow, red, blue, and green, respectively. DNA is shown in purple. **b** Overlay of MM-NLP$_{207W}$ with eNuc (gray) with only 71 bp of DNA, one H2A-H2B dimer, H3–H3′ four-helix bundle, H3′ N-terminal helix, and a single H4 displayed for clarity. Superhelix locations (SHLs) are numbered from 0 to 6 starting at the nucleosome dyad (φ). **c** Comparison of native (light blue) and GraFix (purple) MM-NLP$_{207W}$ electron densities.

histone octamer[34]. In eukaryotes, the 'bowl' (bottom part) of the ladle is shaped by intramolecular interactions and interacts with the H4 C-terminal tail, while the 'handle' extends upwards to interact with H3-αN and H3-α2 helices (Fig. 5b). In Melbournevirus, the 'bowl' is not braced as tightly and the handle is redirected to interact mainly with H3-α2[19]. In Medusavirus, the C-terminal tail of H4 is shorter by three amino acids and is oriented in a way that precludes the formation of a short β-sheet interaction with the 'bowl' of the docking domain, which has even fewer stabilizing intermolecular interactions than Melbournevirus. The handle of the MM-H2A docking domain, which is considerably longer than in either eH2A or Melbournevirus H2A, does not appear to engage in any interactions with H3 and, therefore, is not observed in the density (Fig. 5b).

The histone core is held together by the canonical four-helix bundle interactions between H3–H3′ and between H2B–H4 (Fig. 3b). However, the composition of four-helix bundles in MM-NLP differs from Melbournevirus NLP and eNuc (Fig. 5c, d). In particular, both viral nucleosomes lack the histidine-cysteine configuration in the H3–H3′ four-helix bundle that is typical for nearly all eukaryotic nucleosomes and is hypothesized to convey copper reductase activity (Fig. 5c)[35]. Additionally, both viral nucleosomes lack the close packing of three aromatic rings that are observed in the four-helix bundle formed by eukaryotic H2B–H4. Intriguingly, MM-H2B and MM-H4 each contribute a histidine to the four-helix bundle in a configuration that

resembles that of the eukaryotic H3–H3′ interaction (Fig. 5d, middle and Fig. 5c, left). The high degree of variability in the four-helix bundle regions might serve to avoid the pairing of viral histone dimers with host histones that are likely also present during viral nucleosome assembly.

The charge distribution of the histone octamer surface differs between viral nucleosomes. MM-NLPs exhibit a more pronounced positively charged DNA-interacting ridge compared to eNuc and MV-NLP (Supplementary Fig. 5A). The 'acidic patch', a localized region on each disk "face" of the nucleosome, is a well-established and highly conserved binding site for many chromatin-interacting proteins in eukaryotes[36]. The negative charge in this region of the MM-NLP is less pronounced. Portions of the docking domain that in eNuc and MV-NLP track along the surface of H3–H4 and marginally contribute to the acidic patch are disordered in MM-NLP. However, the missing MM amino acid residues (VGQVAEMAAAAANTG...) are unlikely to contribute to the surface charge in this region. The unique surface features of viral histones, including the MMH2B L1 loop and MM H3 C-terminal extension, further shape the nucleosomal surface (Supplementary Fig. 5B). Notably, the H3 C-terminal tail contributes to forming a pronounced "S" shaped acidic surface along the bottom side of the (H3–H4)$_2$ tetramer that is unique to MM histones (Supplementary Fig. 5C, D). These differences in charge distribution seem to be specific features of MM-NLPs.

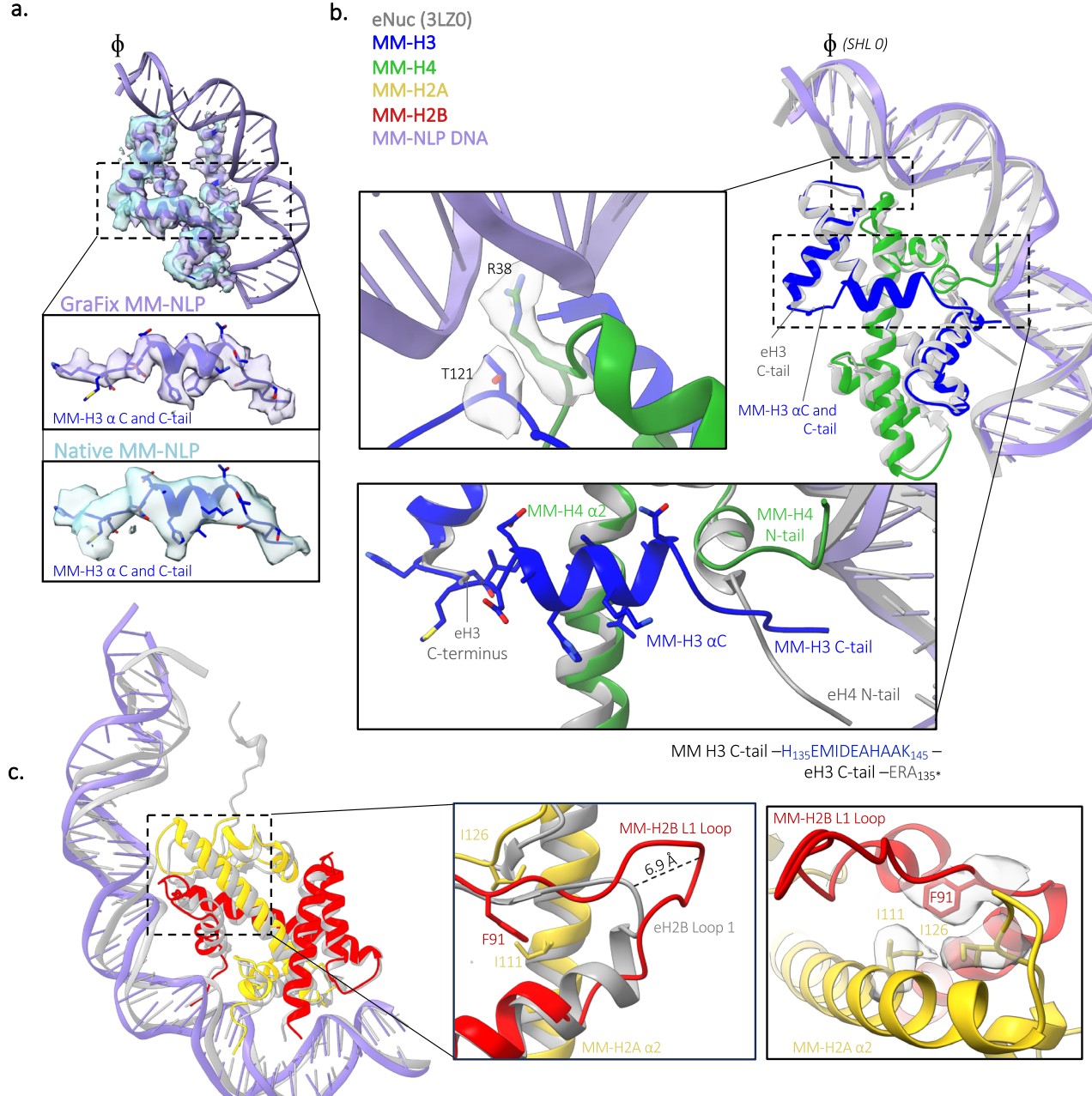

MM H3 C-tail —H$_{135}$EMIDEAHAAK$_{145}$ —
eH3 C-tail —ERA$_{135*}$

**Fig. 4 | Comparison to eNuc reveals unique roles for longer tails and loops.**
**a** Comparison of crosslinked and native MM-NLP. Superposition of GraFix MM-H3 density and native MM-H3 density. The MM-H3 tail from H135 to G153 is displayed (inset), with each density shown as purple and blue surface, respectively.
**b** Superposition of MM-(H3–H4) with eukaryotic (H3–H4) and 30 bp of associated DNA. Close-ups (inset) are provided of (Top) MM residues in DNA minor groove with residue density shown as gray surface, (Bottom) MM-H3 C-tail (blue) orientation in relation to the eukaryotic H3 C-terminus (gray), the eukaryotic (gray), and MM H4 N-tail (green). Corresponding residues of each H3 C-tail are denoted below. **c** Superposition of MM-H2A-H2B dimer aligned with eukaryotic H2A-H2B dimer (gray) and 40 bp of associated DNA. Close-ups (inset) are provided of (left) eH2B (gray) and MM-H2B loop 1, highlighting MM loop extension, (Right) MM-H2B loop 1 and H2A residues form hydrophobic core with density shown as gray surface.

## MM putative linker histone H1 does not compact tri-nucleosomes

Unique amongst histone-encoding NCLDVs, MM has a gene for a putative linker histone H1 (ORF 106) with an uncharacteristically acidic pI of 5, but similar to the pI of H1.1 of its host *A. castellanii* (Fig. 6a and Supplementary Fig. 1E)[22]. The putative MM linker histone H1 (MM-H1) has a low sequence conservation (12.9%) compared to eukaryotic *X. laevis* H1.0. Secondary structure predictions of MM-H1 suggest the presence of the canonical N-terminal winged-helix DNA-binding domain seen in *X. laevis* H1.0 and *A. castellanii* H1.1 (Supplementary

Fig. 1D). Unexpectedly, we predict an additional winged-helix domain in both MM-H1 and in all *A. castellanii* H1 variants, connected by loops that vary in length and sequence, which is supported by AlphaFold models (Fig. 6a and Supplementary Fig. 6). Both winged-helix domains predicted for MM-H1 have a more negative charge compared to *X. laevis* H1.0, with the second predicted domain displaying a predominantly negative charge (Supplementary Fig. 6). Superimposing the predicted structure of MM-H1 with *X. laevis* H1.0 reveals that the four basic amino acids (K40, R42, K52, and R94) required to compact chromatin are not conserved in MM-H1[37]. Instead, the predicted

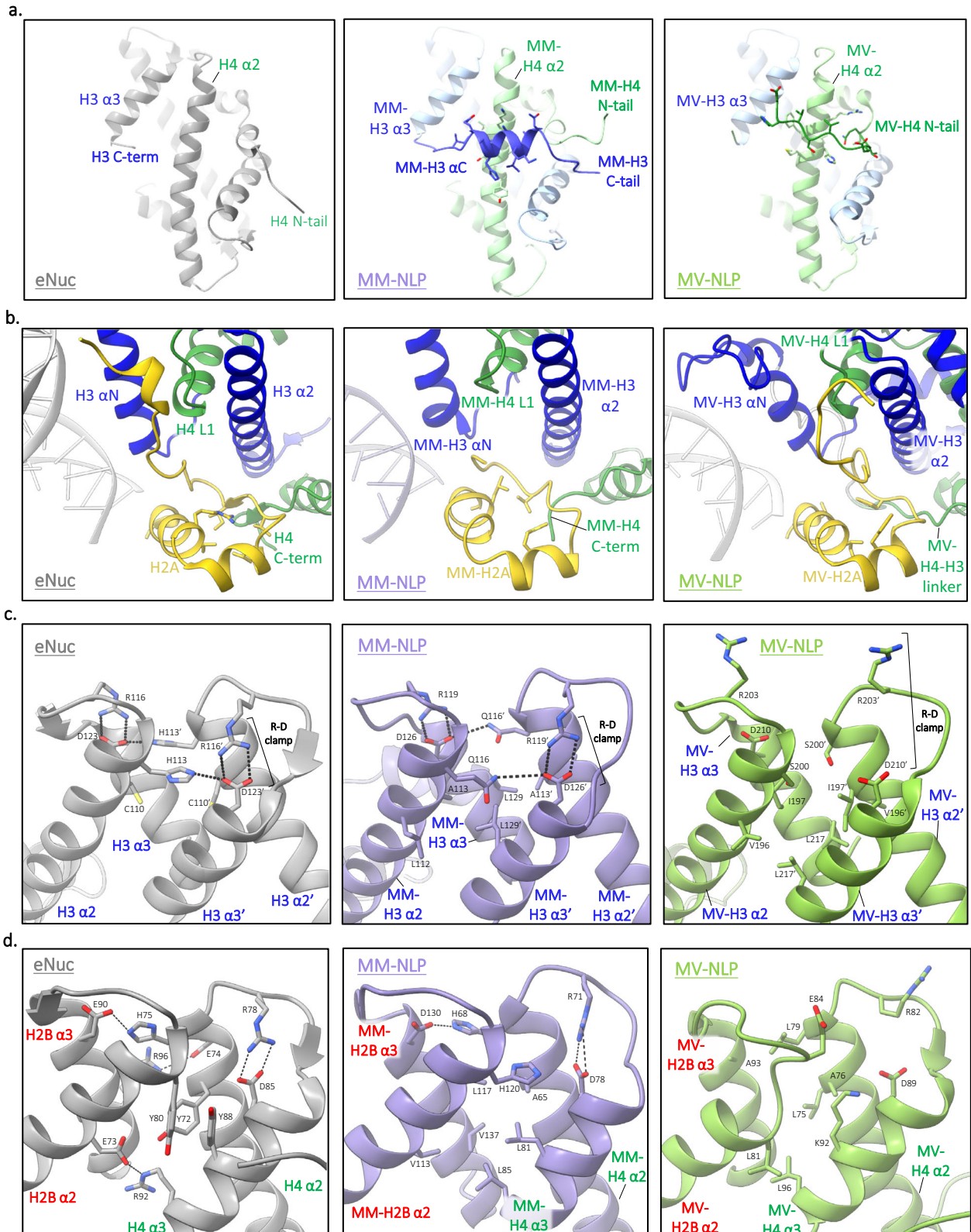

**Fig. 5 | Medusavirus NLPs have unique structural features. a** The H4 α2 helix is stabilized by different structural elements in Medusavirus (MM) and Melbourne-virus (MV) NLPs. H3–H4 heterodimer for eNuc and MM-NLP, and H3–H4 doublet for MV-NLP shown. **b** H2A docking domain (yellow) in eNuc, MM-NLP, MV-NLP with key residues shown as sticks. **c** Residues contributing to the H3–H3′ four-helix bundle interactions in eNuc, MM-NLP, and MV-NLP. **d** Residues contributing to the H2B–H4 four-helix bundle interactions in eNuc, MM-NLP, and MV-NLP.

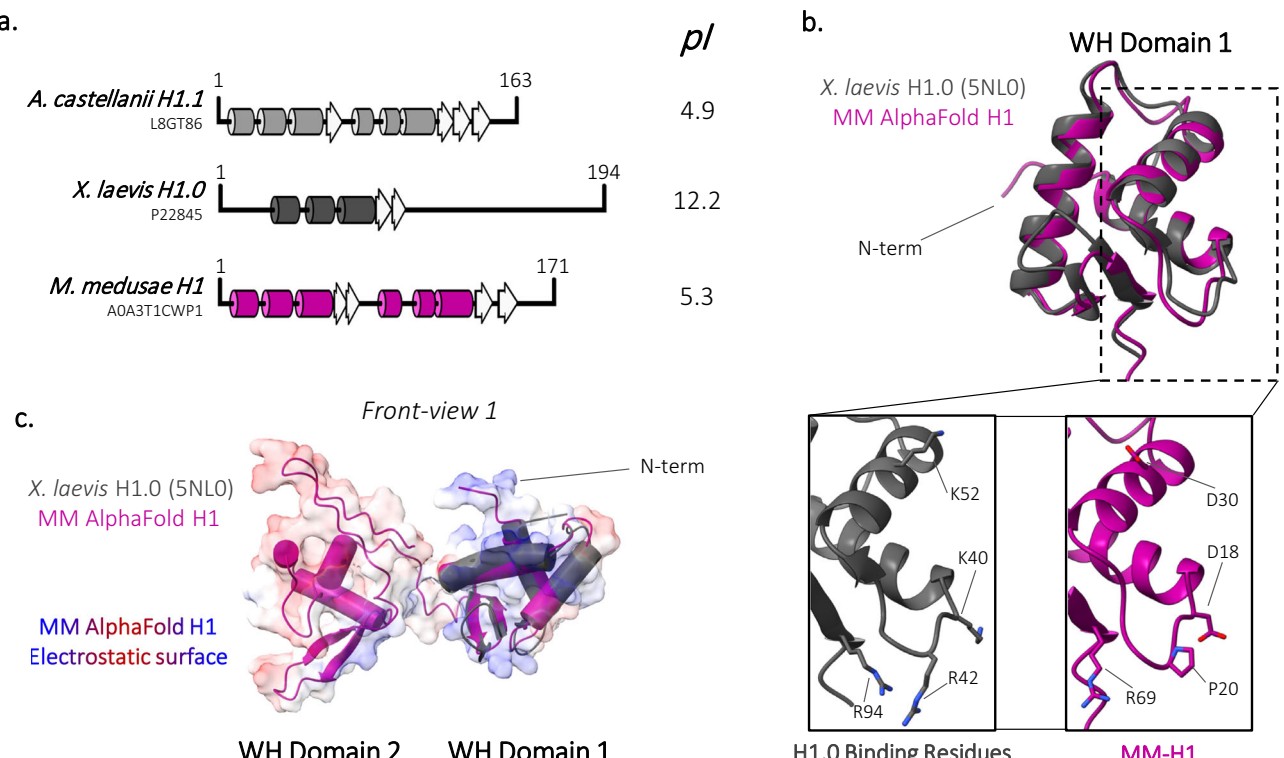

**Fig. 6 | *Medusavirus medusae* linker histone H1 contains a second winged-helix domain and lacks canonical compaction residues. a** *Acanthamoeba castellanii* (*A. castellanii*) H1.1, *Xenopus laevis* (*X. laevis*) H1.0, and *Medusavirus medusae* (MM) linker histones were aligned using HHpred's multiple sequence alignment tool (ClustalΩ). Predicted *A. castellanii* α helices and beta strands in *Mamonoviridae* histones were generated using HHpred's Quick 2D prediction webserver (shown as tubes and arrows, respectively). Known *X. laevis* H1.0 and *A. castellanii* α helices and beta strands representative of the winged-helix (WH) domain shown as tubes and arrows, respectively. Corresponding isoelectric points (pI) of each linker histone are provided to the right. **b** Superposition of AlphaFold MM-H1 (dark pink) and *X. laevis* H1.0 (dark gray; 5NL0) within the charged surface representation of MM-H1. **c** Overlay of MM-H1 (shown in dark pink) and *X. laevis* H1 (dark gray; 5NL0). Inset: *X. laevis* H1 residues involved in chromatin compaction and corresponding residues in MM-H1.

MM-H1 has one equivalent basic residue (R69) and acidic residues (D18, P20, and D30), which suggests MM-H1 may lack the ability to function in compaction (Fig. 6b).

To characterize the putative MM linker histone H1 (MM-H1), we expressed, purified, and refolded the protein for biochemical analysis (Supplementary Fig. 7A). Circular dichroism (CD) of refolded MM-H1 confirms a significant increase in order when compared to *Mus musculus* H1.0 (eH1.0), from ~35% to ~75% (Supplementary Fig. B, C). This is due to an increase in α-helix and β-sheets for MM-H1, as predicted by secondary structure and Alphafold predictions that both suggest the presence of a second winged-helix domain in MM-H1 (Fig. 6c)[38,39].

To test the ability of MM-H1 to bind to DNA, as required if it were to compact chromatin, fluorescence polarization (FP) of labeled 35-mer DNA was monitored at increasing concentrations of H1. While eukaryotic eH1.0 binds to DNA, as is well-established, neither MM-H1 nor Amoeba H1.1 are able to bind (Supplementary Fig. 7D). Using gel shifts, we showed that while eH1 interacts with viral and eukaryotic tri-nucleosomes, MM-H1 does not interact with either (Supplementary Fig. 7E).

AFM has previously been used to demonstrate eNuc-tri compaction by eH1.0, which was characterized by an increase in particle height, as well as by an increase of visually compacted particles[40]. Here, we utilized AFM as an orthogonal assay to probe the ability of the MM-H1 to compact tri-nucleosomes. eNuc-tri compaction was observed with the addition of eH1.0 via an increase in the height distribution of tri-nucleosome particles (Fig. 7a, b, e). This is expected based on structures of H1 bound nucleosome arrays, which show a zig-zag arrangement with non-consecutive nucleosomes 1 and 3 forming a stack, and H1 bound to linker DNA near the dyad of each nucleosome[41].

This stacking likely explains the "bi-lobed" tri-nucleosomes observed via AFM when tri-nucleosomes are incubated with eH1.0. The stacked terminal nucleosomes (nucleosome 1 and 3) have increased height compared to the central nucleosome (nucleosome 2) (Fig. 7a, b, e). The addition of MM-H1 to eNuc-tri did not yield these distinct "bi-lobed" tri-nucleosomes or an increase in particle height (Fig. 7c–e). Compaction was also directly assessed by measuring the total distance between the three nucleosomes in the tri-nucleosomes. Upon the addition of eH1.0 to eNuc-tri there was a significant decrease in distance between nucleosomes that was not seen upon the addition of MM-H1 (Fig. 7f). However, this analysis precludes the ability to measure the most compact tri-nucleosomes where nucleosomes are not individually visible (i.e., bi-lobed), and thus underestimates the level of compaction observed with eH1.0. Using these two AFM analyses, no tri-nucleosome compaction was observed upon the addition of MM-H1. Together this data suggests that MM-H1 does not function in compaction under these conditions.

## Discussion

Once believed to be unique to the eukaryotic domain of life, the universe of histone-encoding organisms continues to expand to now include most archaea, some bacteria, and recently, several members of giant viruses (NCLDV). Structural analysis of non-eukaryotic histone-DNA complexes has demonstrated a remarkable diversity in histone-based DNA organization. For example, bacterial histone dimers bind DNA edge-on and filament around straight DNA, while archaeal histone homodimers wrap variable lengths of DNA into dynamic 'hypernucleosome slinkies'[42–44]. In contrast, giant viruses encode clearly recognizable homologs of the four eukaryotic core histones

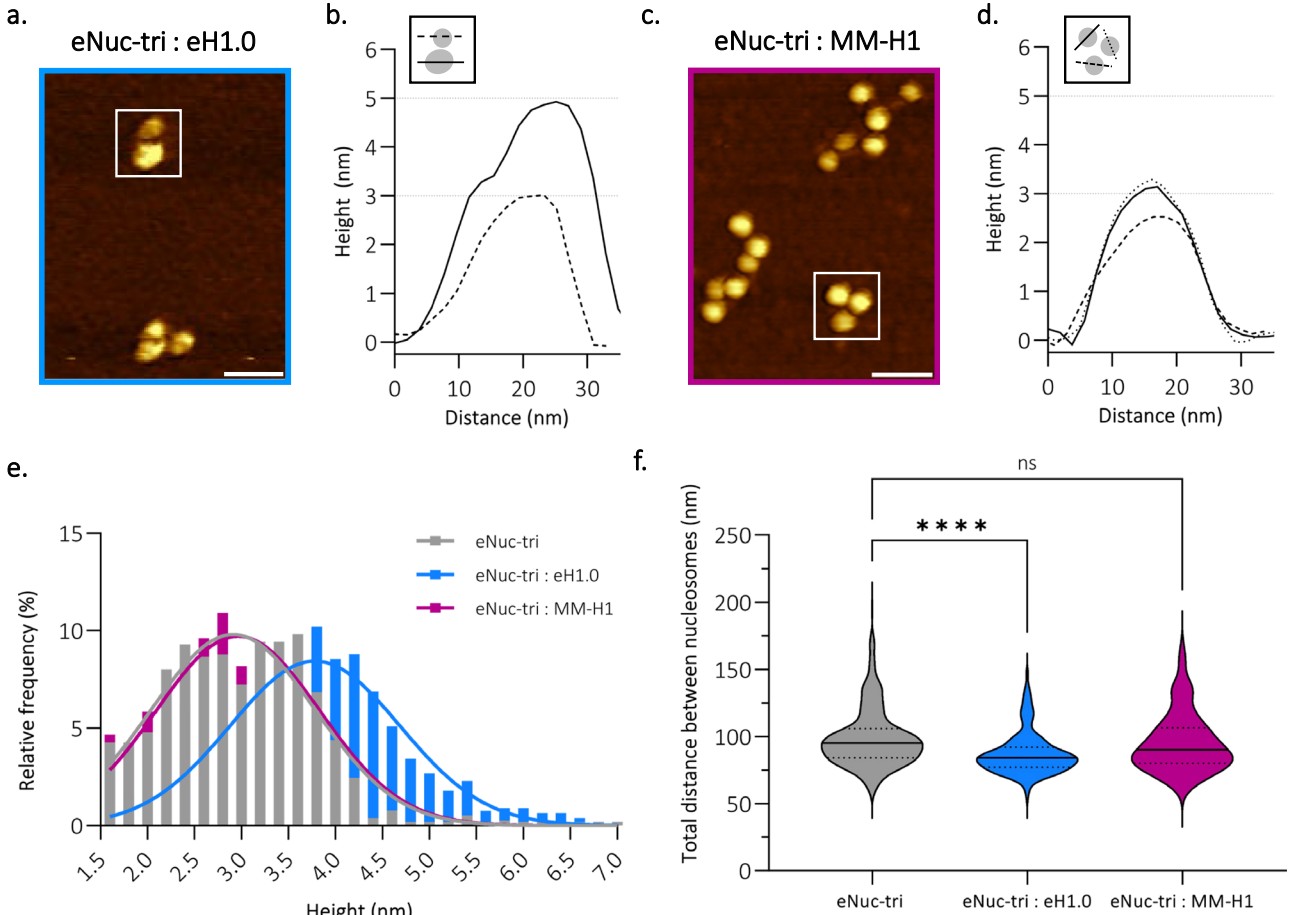

**Fig. 7 | *Medusavirus medusae* linker histone H1 does not compact tri-nucleosomes. a** Representative AFM topography image of eNuc-tri:eH1.0 sample, imaged in air (scale bar = 50 nm). More than three technical replicates of this representative data were observed with same result. **b** Height profile through particles (indicated by white box in **a**) along the lines as depicted in graphical inset. **c** Representative AFM topography image of eNuc-tri:MM-H1 sample, imaged in air (scale bar = 50 nm). More than three technical replicates of this representative data were observed with same result. **d** Height profile through particles (indicated by white box in **c**) along the lines as depicted in graphical inset. Dashed lines at 3 and 5 nm shown for reference. **e** Histogram and Gaussian fitting (data are shown as mean ± SD ($\mu \pm \sigma$)) of particle height for eNuc-tri alone (2.9 ± 0.9 nm, $N = 774$, gray) and after incubation with either eH1.0 (3.8 ± 0.9 nm, $N = 784$, blue) or MM-H1 (3.0 ± 0.9, $N = 770$, pink). **f** AFM compaction analysis for eNuc-tri alone (99 ± 22 nm, $N = 260$, gray) and after incubation with either eH1.0 (87 ± 16 nm, $N = 245$, blue) or MM-H1 (95 ± 22 nm, $N = 249$, pink). These values represent mean ± SD and statistical significance was determined by unpaired two-tailed Student's *t* test, **** represents $P = 4.69 \times 10^{-11}$, ns represents $P = 0.0703$. Center line, median; dashed lines, upper and lower quartiles. Source data are provided as a Source Data file.

(H2A, H2B, H3, and H4). For Melbournevirus, we and others have shown that they assemble into nucleosome-like particles that wrap ~130 bp of DNA[19,21]. Phylogenetic analysis suggests that viral histones diverged prior to the emergence of LECA[22,45–48]. Instances of histone doublets (or even triplets and quadruplets) have been described in NCLDV, which enforces specific pairing of histones and suggests a potential role in the origin of eukaryotic histone dimers/tetramers[2,9,48]. With some NCLDV only encoding one doublet pair, it is possible that NCLDV histones doublets have evolved independent functions in viral genome packaging; although it is unclear if these independent functions were developed before or after complete sets of core histone singlets were encoded in eukaryotes or even MM[47]. MM histones are placed at the root of most histone phylogenetic trees and, thus, add to theories regarding nucleosome evolution[5,12,18,22]. Their resemblance to eukaryotic core histones is plausibly a result of gene exchange through horizontal gene transfer (HGT) post-LECA, however their sequence similarity is low compared to its host *A. castellanii* and other viral histones. As such, the role of MM core histones in the origin of the nucleosome remains a mystery.

We show that key eukaryotic nucleosome features are conserved within MM-NLPs, despite the low conservation of amino acid sequences. This includes the overall geometry and arrangement of the DNA superhelix, which is achieved through the interactions between the DNA minor groove backbone and the main chain of antiparallel L1-L2 loops of the histones, as well as the N-termini of the histone fold α1 helices[1,19,21]. However, MM histones diverge in the lengths of tails and loops within the histone fold, which distinctly shape the nucleosomal surface. For example, the extended MM-H2B loop 1 may act as an 'arm' for other interacting viral or host proteins, including putative nucleosome assembly, remodeling factors, and histone chaperones, and/or may change the way in which nucleosomes pack against each other into higher-order structures. A similar role for an extended loop has been observed for the centromeric histone variant CenpA[49].

The unique extended MM-H3 C-terminal region forms an α-helix that lays across the H3–H4 histone fold. While the H3 C-terminal α helix is unique to MM, its pathway across H4 α2 mimics that of the Melbournevirus H4 N-term tail, which extends out and lays across the histone fold dimer in the exact same location but from the opposite direction. This conserved requirement of a 'stabilizing bracket' for the H3–H4 histone dimer that also reconfigures the nucleosome surface (either from H3 in MM or from H4 in MV) appears to be a feature that is unique to viral nucleosomes. Furthermore, the extended H2B L1 loop

stabilizes the H2A-H2B heterodimer in Medusavirus. As such, it appears that viral histone heterodimers have an additional requirement for stabilization beyond the histone fold (either by additional structure elements as in Medusavirus, or through fusion into histone doublets in Melbournevirus). Whether this is due to the absence of histone chaperones that would otherwise stabilize the histone fold prior to incorporation (none have been identified as of now), or because combination with host histones must be avoided, remains to be seen. The high level of divergence in the regions stabilizing the $(H3–H4)_2$ tetramer and in tethering the $(H3–H4)_2$ to the H2A-H2B dimer (through four-helix bundle interactions and through the H2A docking domain) might have evolved for similar reasons.

Unlike most histone-encoding NCLDV, Medusavirus encodes an acidic putative linker histone H1 that maintains a much higher level of sequence identity to H1 encoded by the host *A. castellanii* than what is observed for the core MM histones. This suggests that the predicted H1 was shared through HGT at a later stage in the evolving host-virus relationship. MM-H1 has the canonical winged-helix domain seen in *X. laevis* H1.0, but only one of the four established residues is required for chromatin compaction. Under our conditions, MM-H1 did not display any DNA or nucleosome binding activity, suggesting that this protein has evolved to fulfill different functions, also suggested by its distinct early expression in the virus life cycle before MM core histones. MM-H1 is not packaged in the virions, which carry the core histones, suggesting that the MM-H1 may function to regulate the host chromatin accessibility[24]. Intriguingly, the Amoeba host contains three putative linker histone genes, all of which have a canonical H1-like winged-helix domain (RMSD of Alphafold models with eH1 is ~3 Å), but vary in the extent of additional predicted domains and disordered regions. The role of any of these linker histones in shaping Amoeba chromatin structure is unknown.

During the assembly of Melbournevirus, viral histones reside exclusively in the viral factories in the host cytoplasm to mature into virions upon Melbournevirus infection[19]. In contrast, Medusavirus maturation relies heavily on the host nucleus. Viral particles are independently produced in the cytoplasm, but viral DNA replication occurs in the nucleus, and only the assembled capsids that are close to the nuclear membrane are filled with DNA. This could allow the opportunity for the exchange of protein and DNA through open membrane gaps of the MM viral particle positioned near the nuclear membrane. Whether the DNA is already assembled into chromatin in the nucleus, during transit into the viral factory, or whether nucleosome assembly takes place in the capsid remains to be determined[50]. The pronounced differences in the four-helix bundle interfaces holding together the $(H3–H4)_2$ tetramer and tethering the $(H3–H4)_2$ to the H2A-H2B dimer might have evolved to prevent the formation of hybrid nucleosomes consisting of a host and virally encoded histones, although this requires further investigation. While histones are essential for Melbournevirus fitness and infectivity, the importance of MM histones for Medusavirus fitness has yet to be explored[19,22]. As more metagenomes are discovered, more organisms will reveal encoded histones to aid in our current narrow understanding of NCLDV chromatin organization.

## Methods

### Histone sequence alignment and secondary structure prediction

Predicted *Mamonoviridae* family and *Marseilleviridae* family core histone-like protein sequences were aligned with eukaryotic histone sequences with HHpred's Multiple Alignment using Fast Fourier Transform (MAFFT) with a 1.52 gap open penalty[22,23]. The sequence similarly and identity of *Medusavirus medusae* (MM), *Medusavirus stheno*, Clandestinovirus, Marseillevirus, and Melbournevirus relative to each core eukaryotic histone were obtained using the Sequence

Manipulation Suite webserver (SMS)[51]. To demonstrate structural conservation of the canonical core histone fold domain between *Eukarya* and *Mamonoviridae*, protein secondary structures were predicted using the MAFFT alignment on the HHpred's Quick 2D structural prediction webserver[52]. The isoelectric point of each protein was provided by protparam.

Predicted *Mamonoviridae* family and Clandestinovirus linker histone-like protein sequences were aligned against host *Acanthamoeba castellanii* H1.1, *Xenopus laevis* H1, and *Gallus gallus* H5) with HHpred's MAFFT with a 1.52 gap open penalty[22,23]. The sequence similarly and identity of MM, Medusavirus stheno, and Clandestinovirus, relative to each linker eukaryotic histone were obtained using the SMS[51]. To demonstrate structural conservation of the canonical winged-helix domain between *Eukarya* and *Mamonoviridae*, protein secondary structures were predicted using the MAFFT alignment on the HHpred's Quick 2D webserver[52].

### MM histone expression, purification, and refolding

MM ORF 318 (H2A), ORF 61 (H2B), ORF 255 (H3), ORF 254 (H4), and ORF 106 (H1) were each cloned into a pET-28a plasmid for expression and purification from *Escherichia coli*. (*E. coli*), utilizing adaptations of well-established eukaryotic histone protocols[26,53]. Expression of each histone was performed in 6 L of Rosetta 2(DE3) *E.coli* cells (Sigma, #71401), with induction of 0.5 mM IPTG at OD:0.4–0.6 and growth at 14 °C for 18 hrs. Inclusion bodies were isolated from the cells utilizing a minor adaptation from published protocols, including a 30 min stir of the cell suspension in wash buffer containing 50 μL of DNase I (NEB, # M0303L), 5 mM $MgCl_2$, 50 μg of Lysozyme (Sigma, # L6876) and 10 mM $CaCl_2$ before tissumizing[26]. Isolated inclusion bodies were treated with 1 mL of 99.7% DMSO for 30 min before stirring with 40 mL of the denaturing lysis buffer (6 M Guanidinium HCL, 20 mM sodium acetate pH 5.2, and 200 mM NaCl) for 30 min. After stirring, samples were tissumized for 25 s intervals at 30% approximately 4–5 times until the viscosity of the lysate resembled water. The lysate was pelleted, and the supernatant was filtered with a 0.4 μM syringe filter.

The filtered lysate was applied to a 5 mL His-Trap HP column in nickel loading buffer (8 M urea, 20 mM sodium acetate pH 5.2, 200 mM NaCl, and 20 mM Imidazole) and eluted utilizing a gradient of the nickel elution buffer (8 M urea, 20 mM sodium acetate pH 5.2, 200 mM NaCl and 1 M Imidazole). Isolated fractions containing each MM core histone (H2A, H2B, H3 and H4) were combined and run over a TSK-SP cation exchange column using the SAUDE 200 buffer (8 M urea, 20 mM sodium acetate pH 5.2, 200 mM NaCl, 5 mM βME and 1 mM EDTA) and eluted using a gradient of the SAUDE 1000 buffer (8 M urea, 20 mM sodium acetate pH 5.2, 1000 mM NaCl, 5 mM βME and 1 mM EDTA). For the putative linker histone H1, fractions from the His-Trap HP column were placed over a Mono-Q anion exchange column in SAUDE buffer. Eluted fractions from the TSK-SP or Mono-Q, containing each individual histone, were combined and dialyzed into 5 mM βME for lyophilization. All histones were lyophilized and stored at −20 °C[53]. Confirmation of each histone protein ID was performed through LC-MS/MS (Jeremy Balsbaugh, UConn).

For protein refolding, putative linker histone H1 was dialyzed alone into refolding buffer (20 mM Tris-HCL pH 7.5, 2 M NaCl, 1 mM EDTA, 1 mM βME) overnight with multiple buffer changes. Precipitated protein was discarded through centrifugation, and the supernatant was concentrated for application over a size-exclusion S200 equilibrated in the refolding buffer. The sample was stored in 20% glycerol at −80 °C. Conversely, the predicted core histones (H2A, H2B, H3, and H4) of MM were refolded at equal molar concentrations together either as an octamer, as dimers (H2A-H2B), or as putative tetramer $(H3–H4)_2$ to remove denaturant by utilizing previously described protocols[53]. Samples eluted off gel filtration at expected volumes and were stored in 20% glycerol at −80 °C.

## MM-NLP reconstitution

The DNA utilized to reconstitute the MM nucleosome-like particle (NLP) in vitro included the Widom 601 DNA at 147 bp (5′-ATCTGA-GAATCCGGTGCCGAGGCCGCTCAA TTGGTCGTAGACAGCTCTAGCAC CGCTTAAACGCACGTACGCGCTGTCCCCCGCGTTTTAACCGCCAAGG-GGATTACTCCCTAGTCTCCAGGCACGTGTCAGATATATACATCCGA-T−3′), 165 bp (5′-ATCGCCAGGCCTGAGAATCCGGTGCCGAGGCCGCTC AATTGGTCGTA GACAGCTCTAGCACCGCTTAAACGCACGTACGCGC TGTCCCCCGCGTTTTAACCGCCAAGGGGATTACTCCCTAGTCTCCAG GCACGTGTCAGATATATACATCCAGGCCTTGTGGAT-3′) and 207 bp (5′-ATCTAATACTAGGACCCTATACGCGGCCGCATCGGAGAATCC CGG TGCCGAGGCCGCTCAATTGGTCGTAGACAGCTCTAGCACCGCTTAAA CGCACGTACGCGCTGTCCCCCGCGTTTTAACCGCCAAGGGGATTACT CCCTAGTCTCCAGGCACGTGTCAGATATATACATCGATTGCATGTGG ATCCGAATTCATATTAATGAT-3′) lengths; along with an in-house 150 bp 50% G/C content (random sequence generated computationally) double-stranded DNA (5′GCTAGTCCGTCTTCTACTCTGAAAT GAGCAGTCCTAGTCAGCAAGATCGCTCAGCCAACTTTCTACCAGCGC AACCCTAATCTACCCCATGAATGAAGCCGCACCCAAAACCGCATTC-TAAGGAGTGACATTAACCCTCGGTGAGGATGT 3′)[27]. MM octamer and DNA were mixed at a range of ratios of DNA to octamer and reconstituted by salt-gradient dialysis from 2 M to 50 mM NaCl (ideal ratio 1.0: 1.6). All eukaryotic histones (*Xenopus laevis*) utilized in the reconstitution of eNuc controls were supplied by the Histone Source (Hataichanok Scherman, Colorado State University). Reconstituted samples were analyzed by 5% native-PAGE.

## Sedimentation velocity analytical ultracentrifugation

To evaluate the homogeneity, molecular size, and molecular shape of MM-NLP$_{207W}$; we used SV-AUC with absorbance optics ($\lambda = 280$ nM). MM-NLP$_{207W}$ at 250 nM, was spun at 30,000–35,000 rpm at 20 °C in the Beckman XL-A ultracentrifuge using the An60Ti rotor (50 mM NaCl, 20 mM Tris-HCL, 1 mM EDTA, pH 7.5 and 1 mM DTT). Sedimentation velocity data analysis were performed in UltraScan III version 4.0 to determine sedimentation coefficients (S$_{(20,W)}$), frictional ratios (f/f$_0$) and molecular weights of each sample and containing populations using established protocols[19,54,55]. Integral S$_{(20,W)}$ distributions plots are displayed using GraphPad Prism version 10.0.0 for Windows, GraphPad Software, Boston, Massachusetts USA, www.graphpad.com.

## Thermal stability assay

MM-NLPs and eNucs were reconstituted on Widom 601 at 207 bp and Random 150 bp DNA as described above. Melbournevirus NLPs were also reconstituted on Widom 601 at 207 bp DNA as described above[19]. Two replicates of each nucleosome (800 nM) were incubated in the presence of SYPRO Orange for 1 min at 25 °C, in 20 mM Tris-HCL (pH 7.5), containing 5 mM DTT and 50 mM NaCl. SYPRO Orange is provided SIGMA-ALDRICH as a 5000× concentrated solution (Catalog #: S5692). Each reaction was performed in a final volume of 20 μL, containing 2.5 μL of 62.5-fold diluted SYPRO Orange 5000x (final concentration 8×). Samples were increased by 1 °C and maintained for 1 min at the increased temperature. Fluorescence measurements were measured using a StepOnePlus Real-Time PCR unit (Applied Biosystems) after each 1 min incubation step. This sequential process was repeated for each 1 °C from 25 °C to 95 °C measuring using the Cal Orange filter (fluorescence emission maximum, 560 nm). Normalized plots displaying the relative fluorescence release of SYPRO Orange over 25 °C to 95 °C are displayed using GraphPad Prism version 10.0.0 for Windows, GraphPad Software, Boston, Massachusetts USA, www.graphpad.com. The thermal melting point (Tm) ranges of each NLP sample ($n = 2$) measured in the thermal stability assay were determined by locating the temperature at the lowest point of the fluorescence derivative (−d/dU RFU).

## Sucrose gradient sedimentation and GraFix crosslinking

A continuous 10-30% (w/v) sucrose gradient was prepared in a 13.2 mL Beckman polypropylene coulter tube (#331372). To form the gradient, 6 mL of the top solution (50 mM NaCl, 20 mM HEPES, 1 mM EDTA, pH 7.5 and 10% sucrose) was added to the coulter tube and then 6 mL of the bottom solution (50 mM NaCl, 20 mM HEPES, 1 mM EDTA, pH 7.5 and 30% sucrose) was slowly added from the bottom of the tube, pushing the top solution up until the demarcation line reached halfway. A gradient maker (BioComp Gradient Master) was then used to generate the continuous gradient. A 200 μL MM-NLP sample (3 μM) was then loaded on top of the gradient and spun at 4 °C for 18 hr at 30,000 rpm (160,000 × $g$) (Beckman, Rotor SW-40Ti). Samples were fractionated with absorbance optics ($\lambda = 260$ nm) for identification of complexes (BioComp Gradient Master). Desired complex fractions were determined using 5% native-PAGE and SDS-PAGE (BioRad, #4569036) to be combined and dialyzed to remove sucrose (50 mM NaCl, 20 mM Tri-HCL, 1 mM EDTA, pH 7.5, and 1 mM DTT). GraFix was performed utilizing the same procedure, with the addition 0.15% glutaraldehyde in the bottom solution to form a continuous gradient with crosslinker. After centrifugation, fractions were dialyzed to remove sucrose and quench the crosslinking reaction (50 mM NaCl, 20 mM Tri-HCL, 1 mM EDTA, pH 7.5, and 1 mM DTT).

## Homology modeling

Initial homology modeling of the MM histones ORF 318 (H2A), ORF 61 (H2B), ORF 255 (H3), and ORF 254 (H4) were constructed using SWISSMODEL, with *Xenopus laevis* histones in the context of a nucleosome (1AOI) as a reference[56–58]. To identify steric clashes in the initial model, CPPTRAJ of the Amber MD package (v18) with a cutoff distance of 0.8 Å between over-lapping atoms was utilized[59]. Each clash was manually addressed by modifying rotamers to reduce overlap and to achieve energy minimization in Chimera (v1.14) using default settings[60–62]. This initial MM nucleosome conformation was further refined by fitting into final 3D electron maps as described below.

## Single-particle cryo-EM data collection and processing

**GraFix MM-NLP$_{207}$ cryo-EM.** MM-NLP$_{207}$ particles were isolated from GraFix and concentrated to 1.27 μM using Amicon Ultra-4 centrifugal filters (Ultracel 30 K, Millipore, #UFC810024). C-flat 1.2/1.3 (Cu) grids (Electron Microscopy Sciences, #CF313-50) were glow discharged (Tergeo-EM Plasma Cleaner) at 40 mA for 30 s. 4 μL of diluted sample (0.635 μM) was applied to the grid, blotted (2 s, blot force 6) and plunge frozen in liquid ethane using the VitroBot Mark IV set to 100% humidity at 4 °C. Micrographs were acquired at a nominal ×magnification of 130,000 on a Titan Krios G3i outfitted with a Falcon 4 direct detection camera, using Thermofisher EPU (v3.0). The raw pixel size was 0.97 Å, and 8534 movies were captured with a dose of 50 e−/Å². The defocus range was −0.6 μm to −1.7 μm.

Data processing was performed in cryoSPARC (v4.5.1). Initial processing (motion correction, CTF correction, and micrograph curation) was performed prior to particle picking with blob picker. Particles were extracted and binned 2× for particle curation. Particles were curated via iterative 2D classification and multi-class ab initio reconstructions to yield 159,734 particles that were un-binned for further refinement. Following ab initio reconstructions with un-binned particles, homogenous refinements, non-uniform refinements, CTF refinements, and local refinements were performed. The resulting map was sharpened using cryoSPARC's sharpening tool, and the resolution was obtained using the gold-standard FSC validation tool (Supplementary Fig. 4A).

**Native MM-NLP$_{207}$ cryo-EM.** MM-NLP particles isolated from sucrose gradient sedimentation were concentrated to 2 μM using Amicon

Ultra-4 centrifugal filters (Ultracel 30 K, Millipore, #UFC810024). C-Flat 1.2/1.3 (Cu) grids (Electron Microscopy Sciences, #CF313-50) were glow discharged (Tergeo-EM Plasma Cleaner) at 40 mA for 30 s before 4 μL of sample was applied to the grid and plunged into ethane using the Vitrobot Mark IV at 100% humidity at 4 °C with no wait time, a 2 s blot, and a blot force of 5. Micrographs were acquired with a nominal magnification of 130000x on a Titan Krios G3i (300 kV) outfitted with a Gatan K3 direct detection camera. The raw pixel size was 1.017 Å with movies captured maintaining an electron dose of 46.29 e/Å². The defocus range was −0.8 to −2.2 μm.

4040 movies were processed initially by cryoSPARC (v2.12.4) through motion correction and CTF estimation[63,64]. Exposures were curated to exclude sub-optimal characteristics by inspecting the CTF Fit resolution (Å), relative ice thickness, and defocus range. Approximately 1000 particles were manually picked from the curated exposures to generate a picking model through Topaz (downsampling = 16). Topaz picking yielded 72,768 particles, and these particles were also subjected to two iterative rounds of 2D classification to remove undesirable particles. This yielded 72,768 particles, which were utilized to generate ab initio model that was similarly subjected to homogeneous refinement, non-uniform refinement, and local refinement to improve resolution (Supplementary Fig. 4B).

Comparison between the two maps was conducted using ChimeraX (v1.4). The Fit in map feature was used to generate a cross correlation coefficient for the crosslinked and native maps at their recommended contour levels, 0.09 and 0.118, respectively.

## Model building and refinement

The MM-NLP homology model (generated as described above) was rigid-body docked into the map using ChimeraX (v1.4). The model was then iteratively refined in PHENIX (v.1.21.1-5286) using secondary structure restraints and manual adjustments made to the side chain conformations in Coot (v0.9.8.7). PHENIX (v1.21.1-5286) was used to validate the final model prior to deposition. The model coordinates for the crosslinked MM-NLP were deposited into the protein data bank (PDB) under accession code 8UA7. The cryo-EM maps for the crosslinked and native MM-NLP were deposited into the electron microscopy data bank under accession numbers EMD-42053 and EMD-45981 respectively. All figures of the cryo-EM maps and models were generated using UCSF ChimeraX (v1.4).

## Structural characterization of MM nucleosomes

Comparisons between the MM nucleosome (8UA7), Melbournevirus nucleosome (7N8N), and eukaryotic nucleosome (3LZ0) were conducted using ChimeraX (v1.4)[65]. Structures were aligned with the matchmaker tool in ChimeraX (v1.4) using the best-aligning pair of protein chains between the reference and matching structure. All residue and feature comparisons were determined by previously identified canonical eukaryotic histones features and figures were rendered with ChimeraX (v1.4)[1,65].

## Structural prediction of putative linker histone MM-H1

The sequence of the putative linker histone MM-H1 was folded using AlphaFold (v.2.3.2)[66]. The highest-ranked model was then aligned to the *X. laevis* linker histone H1 in PDB 5NL0 using ChimeraX (v1.4).

## Fluorescence polarization

FP was monitored using a 5′-fluorescein label (6-FAM, represented with an *) on a 25-mer DNA substrate (IDT; 5′-*GCAGCTGGCACGA-CAGGTATGAATC). DNA, held constant at a final concentration of 10 nM, was mixed with the respective H1 protein (0 to 10 μM) in a binding buffer of 20 mM Tris (pH 7.5), 1 mM DTT, and 1 mM EDTA. Samples were incubated at room temperature for 1 hour, followed by measurement of FP in a BMG Labtech CLARIOstar microplate reader.

Data were plotted as a function of H1 concentration in GraphPad (v10.0.0) and fit to equation: $Y = Y_{min} + (Y_{max} - Y_{min}) * X / (EC_{50} + X)$, where $Y_{min}$ and $Y_{max}$ are the minimum and maximum signals, respectively, $EC_{50}$ is the experimental binding constant, and $X$ is the H1 concentration. Each data point represents an average of three technical replicates with standard deviation shown as error bars.

## MM-tri-NLP reconstitution

Tri-NLP reconstitution was performed using the Linker Ended Widom 601-207 bp x3 repeat DNA (LE-Tri) (5′-ATCTAATACTAGGACCCTA-TACGCGGCCGCATCGGAGAATC CCGGTGCCGAGGCCGCTCAATTGG TCGTAGACAGCTCTAGCACCGCTTAAACGCACGTACGCGCTGTCCCC CGCGTTTTAACCGCCAAGGGGATTACTCCCTAGTCTCCAGGCACGTG TCAGATATATACATCGATTGCATGTGGATCCGAATTCATATTAATCAT ATCTAATACTAGGACCCTATACGCGGCCGCATCGGAGAATCCCGGTG CCGAGGCCGCTCAATTGGTCGTAGACAGCTCTAGCACCGCTTAAACG CACGTACGCGCTGTCCCCCGCGTTTTAACCGCCAAGGGGATTACTCC CTAGTCTCCAGGCACGTGTCAGATATATACATCGATTGCATGTGGAT CCGAATTCATATTAATCATATCTAATACTAGGACCCTATACGCGGCCG CATCGGAGAATCCCGGTGCCGAGGCCGCTCAATTGGTCGTAGACAGC TCTAGCACCGCTTAAACGCACGTACGCGCTGTCCCCCGCGTTTTAAC CGCCAAGGGGATTACTCCCTAGTCTCCAGGCACGTGTCAGATATATA-CATCGATTGCATGTGGATCCGAATTCATATTAATGAT-3′), and an in-house generated 500 bp 50% G/C content double stranded DNA (5′-GCTAGTCCGT CTTCTACTCTGAAATGAGCAGTCCTAGTCAGCAAGAT CGCTCAGCCAACTTTCTACCAGCGCAACCCTAATCTACCCCATGAAT-GAAGCCGCACCCAAAACCGCATTCTAAGGAGTGACATTAACCCTCGG TGAGGATGTCCATACAAGCACCTCCTACTACGGATCGAACCGTTAGT TCCCCAACTAAGTCCAAACCGTTAGACCGCTTTCCGTACCATTCCGG TACTTATCTTCGCCACAACCTGAGACAATCCCAAGCTTAAGGCTCGA-CACAGACTGACGAAGGATATATCTCGCCCTAACCGTACCTCTATACC GCCATGAAGGAAGTGCCAAGTAGCCACAGAACCTTGGGATAGCAAGA CTCTATGTCCCAGACCTCACTAACACCGAAGGAAAGTACCCACACAG ACATCAGGAAAACCCTCTGACCACTACGGCGAATGAAAAGTCCAGA GGACCAATACGTTACAGAGGCGACTGGATGT-3′)[27]. MM octamer and DNA were mixed at a 1.0: 4.8 ratio of DNA to octamer and reconstituted into tri-nucleosomes by gradient dialysis[26]. All eukaryotic histones (*Xenopus laevis*) utilized in reconstitution of LE-tri controls were supplied by the Histone Source (Hataichanok Scherman, Colorado State University). Reconstituted samples were tested for quality using 4% native-PAGE and mass photometry to confirm a homogenous sample.

## Mass photometry

Mass photometry measurements were performed on a Refeyn TwoMP mass photometer (Refeyn Ltd). Glass coverslips were first cleaned with isopropanol, deionized water, and dried with $N_2$ gas, before being coated with a 0.01% Poly-L-Lysine solution (Sigma, #P4707) for 20 seconds, rinsed with water, and dried with $N_2$ gas. To form a sample chamber, self-adhesive silicon gaskets were adhered to the top of the treated coverslip. For each measurement, the coverslip was placed on the oil-immersion objective lens, centered on a single well, and 13.5 μl of sample buffer (20 mM HEPES, pH 7.5, 100 mM KCl) was added to the well, and the focal position of the glass surface was determined and held constant using an autofocus system. Nucleosome samples were first diluted to 100–200 nM, before a final 10-fold dilution onto the sample stage (final concentration of 10–20 nM). All dilutions were performed at room temperature in sample buffer (20 mM HEPES, pH 7.5, 100 mM KCl). A 60 s video was recorded immediately after the final dilution, using Refeyn AcquireMP (v2.3.0). A fresh well and dilution were used for each measurement and repeated at least three times for each sample. Tri-nucleosomes were diluted in a buffer so that the number of detected events (particle counts) during the 60 s measurement was ~4000–9000 for optimum data acquisition and processing. A known mass standard (β-amylase and thyroglobulin) was

used to convert image contrast-signal into mass units. To calculate the molecular weight of the main species observed on the particle counts versus molecular mass distribution histograms, we used the Gaussian function in the DiscoverMP software (v2.3.0).

## Atomic force microscopy

MM-tri-NLP and eNuc-tri were reconstituted via salt-gradient dialysis as described above[40]. AFM slides were prepared by freshly cleaving the mica, treated with APTES for 30 minutes, rinsed with water, and then dried with nitrogen gas utilizing a 0.22 μm PES filter. Tri-nucleosome samples were diluted in 20 mM Tris-HCl pH 7.5 and 1 mM EDTA to a final concentration of 1 or 2 ng/μl. Immediately after dilution, samples were applied to the APTES mica slide for 2 minutes, rinsed with water, and dried with filtered $N_2$ gas. For H1 samples, the respective tri-nucleosome was incubated with H1 in a 1:4 ratio for 30 minutes at room temperature, before being diluted two-fold and deposited on the APTES surface (as described above). All samples were imaged in the air on JPK/Bruker NanoWizard 3.0 using JPK/Bruker SPM software (v6.4.22) with TAP300-Gold (Ted Pella, # TAP300GD-G-10) cantilevers. Images were collected at a scan size of 1 × 1 or 2 × 2 μm, a resolution of 512 × 512 pixels, and at 1–3 Hz.

AFM data was leveled, processed, and analyzed using the Gwyddion software (v2.65)[67]. For particle height analysis, a mask at 1.5 nm was applied to identify particles, followed by extraction of the maximum height for each particle. Rare particles (>8 nm) were excluded as debris or aggregation. For compaction analysis, the total distance between nucleosomes was measured in Image J (v1.54) by manually measuring the perimeter of a triangle formed between the three nucleosomes. Of note, tri-nucleosome particles for compaction analysis were manually picked and only tri-nucleosomes with three visible nucleosomes were included in this analysis. Analysis and graphing of extracted data was completed in GraphPad Prism (v10.0.0, GraphPad Software, Boston, Massachusetts USA, www.graphpad.com).

## Circular dichroism

Far-UV CD spectra of H1 proteins (0.5 mg/ml) were recorded at room temperature on a ChirascanPlus (Applied Photophysics Ltd, UK) spectrometer using quartz tubes (0.5 mm optical path length) using CHIRASCAN software (v4.7). The measurements were recorded in the 180–260 nm wavelength range with a 0.5 nm step size. All experiments were carried out in 0.02 M Sodium Phosphate and 0.2 M Sodium Fluoride at pH 7.5. For each H1 protein, five replicate CD spectra were averaged, baseline-corrected for signal contributions by the buffer. For secondary structure analysis of CD spectra, the DichroIDP program (v1.0.2) was used[68]. This CD analysis program was chosen as it is suitable for analyses of proteins containing significant amounts of disordered structures (as seen in eH1.0). Data was graphed using GraphPad Prism (v10.0.0, GraphPad Software, Boston, MA USA, www.graphpad.com).

## Reporting summary

Further information on research design is available in the Nature Portfolio Reporting Summary linked to this article.

## Data availability

The AFM data generated in this study have been deposited in the Figshare database [https://doi.org/10.6084/m9.figshare.c.7454029.v1]. Cryo-EM density maps have been deposited in the Electron Microscopy Data Bank under the accession numbers EMD-42053 (3.3 Å MM-NLP), EMD-45981 (Native MM-NLP). Model coordinates have been deposited into the PDB under accession number 8UA7 (3.3 Å MM-NLP). Atomic coordinates from previously published X-ray crystal and Cryo-EM structures are available from the PDB: 3LZ0 (eNuc), 7N8N (MV-NLP), 5NL0 (X. laevis H1.0). Source data are provided with this paper.

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

## Acknowledgements

We thank Charles Moe (previously CU Boulder) and Dr. Erik Hartwick for their help with data collection at the Biochemistry Krios Electron Microscopy Facility (BioKEM) at CU Boulder (RRID: SCR_019057). We thank the Shared Instruments Pool (SIP) core facility (RRID: SCR_018986), Department of Biochemistry, University of Colorado Boulder, and Dr. Annette Erbse for the use of the shared research instrumentation infrastructure. The software used in this project was

curated by SBGrid. This work utilized the CUmulus on-premise cloud service at the University of Colorado Boulder. Cumulus is jointly funded by the National Science Foundation (award OAC-1925766) and the University of Colorado Boulder. This work utilized the Blanca condo computing resource at the University of Colorado Boulder. Blanca is jointly funded by computing users and the University of Colorado Boulder. Data storage is supported by the University of Colorado Boulder PetaLibrary. We thank Dr. Shawn Laursen for the development of the 50% G/C 150 bp sequence DNA utilized in thermal assays. N.M.H. is a Howard Hughes Medical Institute Fellow of the Damon Runyon Cancer Research Foundation (DRG-2499-23, N.M.H.). S.W. is a trainee in the Signaling and Cellular Regulation T32 training program (T32 GM142607, S.W.). Funded by the Howard Hughes Medical Institute (K.L.).

## Author contributions

C.M.T. conceptualization; methodology; investigation; data curation; validation; formal analysis; writing (original draft, review, and editing); visualization. N.M.H. methodology; investigation; data curation; validation; formal analysis; writing (original draft, review, and editing); visualization. S.W. investigation; data curation; validation; formal analysis; writing (review and editing); visualization. K.L. conceptualization; funding acquisition; project administration; supervision; writing (original draft, review, and editing).

## Competing interests

The authors declare no competing interests.
