## [Transparent Peer Review file · Nature Communications]

Characterization of Medusavirus encoded histones reveals nucleosome-like structures and a unique linker histone

Corresponding Author: Professor Karolin Luger

Version 0:

Reviewer comments:

Reviewer #1

(Remarks to the Author)

The manuscript "Characterization of Medusavirus encoded histones reveals nucleosome-like structures and a unique linker histone" (NCOMMS-24-22386) by Toner et al. reports the structure of the Medusavirus medusae (MM) nucleosome, which is known to encode the four core histones and linker histone H1 on separate genes. In this manuscript, the authors reconstituted the nucleosome-like structure of Medusavirus, a giant DNA virus, with its four histones and an artificial positioning sequence, determined the structure by cryo-EM at a resolution of 4.3 Å revealing a unique viral nucleosome-like structure. Nucleosome arrays were then reconstituted, and AFM was used to observe the formation of "beads-on-a-string" structures. In addition to the structural results, biochemical analyses using analytical ultracentrifugation and thermal stability assays, as well as reconstitution of the viral nucleosome using "random" sequences that are not strong positioning sequences, indicate that the nucleosome-like structures formed by viral histones are unstable compared with those of humans.

This manuscript presents a pioneering study of viral nucleosome-like structures and their properties, which are still poorly understood. However, the author's assertions derived from cryo-EM results are poorly supported due to the low resolution of the cryo-structure. Discussion of the interaction of amino acid side chains despite the low resolution seems somewhat hazardous. The data are also weak in terms of novelty and impact. Above all, there is a lack of data on the role that these nucleosome-like structures might play in viruses. Therefore, the manuscript may be unsuitable for publication in Nature Communications.

The authors are strongly encouraged to respond to the following comments to improve the overall quality of the manuscript.

Major comments:

1. The resolution of the determined structure may not be high enough to assess the directions of the amino acid side chains discussed in Figures 4B and 4C, 5B, 5C and 5D. The authors should improve the quality of the maps to confirm their discussion. Alternatively, for example for Figures 4B and 5B, where the interaction between H4 and the C-tail of H3 is analyzed, the authors could perform a mutant assay to validate their assertion.
2. The authors compare the structure of the Medusavirus nucleosome with the Melbournevirus nucleosome previously determined and point out that the Medusavirus nucleosome is closer to the eukaryotic nucleosome. However, I don't entirely agree with this affirmation. The structural differences are not obvious and, what's more, are similar in several respects. The authors should focus on the functional differences, which could significantly strengthen their claim.
3. Medusavirus medusae is a unique virus that possesses a binding histone H1, absent in other NCLDVs. The authors showed by AFM imaging that Medusavirus H1 lacks the ability to compact chromatin. As the authors show, Medusavirus H1 does not appear to be able to compact eukaryotic tri-nucleosomes. However, the function of Medusavirus H1 should be further analysed by structural analyses involving Medusavirus tri-nucleosome. It would also be interesting to know whether acidic Medusavirus H1 could function as a viral chromatin protein; by performing EMSA or other methods, can purified Medusavirus H1 bind to viral nucleosome-like structures? the authors should show whether Medusavirus H1's inability to bind to nucleosomes explains its inability to function as a compaction protein. From a general point of view, the authors should pursue the analysis of the Medusavirus H1-bound nucleosome in parallel with other biochemical experiments, and not only by discussing the cryo-EM structure they have determined, to fully understand the function of this viral nucleosome and its putative H1 histone.

Minor comments:

1: P21, Sedimentation velocity analytical ultracentrifugation (SV-AUC):

In the text, it is cited that “To evaluate the homogeneity, molecular size, and molecular shape of MM-NLPs and 79 MM-LE-tri-NLP in solution; we used SV-AUC with absorbance optics ($\lambda = 280$ nm). MM-NLP 80 of various DNA lengths and MM-tri-NLP, all at 250 nM, were spun at 30,000 – 35,000 rpm at 81 20°C in the Beckman XL-A ultracentrifuge” However, the analytical ultracentrifuge data in the manuscript appear to be based on 207W DNA. Please correct.

2: P24 L142-144

Please check that the model's name and pixel size of the cryo-EM are correct.

Please also correct the defocus range “-2.2 μM ” to “ μm ”.

3: Please correct the following typos:

- P8 L48: “SYPRO Orange” and not “SYBRO Orange”.

- P24 L156: “ab initio model” and not “an initio model”.

Reviewer #2

(Remarks to the Author)

In this paper, the authors present the structure of the nucleosome from *Medusavirus medusae* (MM), which bears a resemblance to the canonical nucleosome but with certain distinct features. The resolution of the structure is somewhat limited, and the authors themselves acknowledge that some of the differences observed from the canonical nucleosome might be due to their inability to construct parts of the viral nucleosome. This limits the overall quality and reliability of the data. The study does not discuss biological function and significance of this viral nucleosome, which restricts the novelty of the research. In my view, the current state of the work may not meet the publication standards of *Nature Communications*.

1. At one point, the authors mention an acidic patch, stating: “The negative charge in this region of the MM-NLP is less pronounced (Figure 4D), possibly due to our inability to construct parts of the H2A docking domain.” This raises concerns about the quality of the model and the potential impact on other regions. The authors should improve their structural analysis to obtain a higher resolution structure, enabling them to better build this and other parts of the nucleosome.

This also pertains to Figures 4 and 5, where the authors discuss interactions between unique tails and loops in MM-NLP and interactions between the four-helix bundle (which is crucial for nucleosome stability) at a 4.3Å resolution structure, where side chains are not fully resolved. The authors need to demonstrate how the regions they discuss in the main figures fit into the experimental density.

The authors should improve their structural analysis and the resolution of their maps to build a more reliable model.

2. The authors demonstrate that the putative linker histone h1 does not compact tri-nucleosomes due to the absence of crucial compaction residues and an acidic electrostatic surface. It would be beneficial if the authors could demonstrate whether this histone-like protein binds to chromatin both *in vitro* and *in vivo*. From their experiments, it's not even apparent if this linker histone-like protein can bind to the viral nucleosome *in vitro*.

3. In Table 1, the experimental molecular weight for MM-NLP displays a large range (238-343). Considering the authors' data indicates low stability of the octamer (Figure S3) and nucleosome (Figure S3, Figure 2), did the authors consider different possibilities for the observed experimental molecular weight of the sample? It's possible that their sample could be a mixture of nucleosomes, hexasomes, or even tetrasomes.

4. The authors suggest the Viral Eukaryogenesis hypothesis in several parts of the text, which is purely speculative and the data presented by the authors do not support this hypothesis any more than they do horizontal gene transfer. The data presented in this manuscript are insufficient to support any evolutionary theory.

5. Could the authors elaborate on how this MM-NLP differs from other NCLDV NLPs?

Minor points:

1. Would it be possible for the authors to display the complete MM-NLP structure in the same orientation as in Figure 4E? This would allow for a better understanding of the contribution of the acidic H3 C-tail.

2. The authors propose that the N-terminal tail of H4 in Melbournevirus performs the same function as the MM H3 C-tail. Does the Melbournevirus H4 N-tail exhibit a similar charged surface as depicted for the MM-H3 tail?

3. The authors mention that *A. castellanii* possesses three linker histones. Is there any available data demonstrating that these proteins bind to chromatin *in vivo*?

4. Could the authors provide a clearer explanation of the connection between the H3 helix and the tethering of the H2AH2B dimer to the h3H4 tetramer? The current explanation is somewhat confusing.

5. Would it be possible for the authors to indicate the location of the H3-H3' and H2B-H4 four-helix bundle on the model of the nucleosome structure in Figure 5, to help the reader's orientation (as they did in Figure 4)?

Reviewer #3

(Remarks to the Author)

This article from Toner et al presents the structure of nucleosome-like proteins from the virus Medusavirus medusae, highlighting differences between the Medusavirus nucleosomes, eukaryotic nucleosomes and nucleosomes from the Melbournevirus genus of viruses. This article is generally well-written, and the conclusions of the article are supported by the data presented. It is unclear how one of the major conclusions of the paper, that Medusavirus NLPs are structurally more similar to eukaryotic nucleosomes, is reached, so this could be better clarified in the article, particularly as it appears to be key to the main message of the article. The results overall provide new structural insights into nucleosome/histone evolution, and the article is suitable for publication in Nature Communications once the below suggested changes are addressed.

My major concerns are:

- Please deposit the lower-resolution un-fixed map in the EMDB and include the deposition code in the manuscript, as this map was discussed extensively in the manuscript.
- Line 62, page 6 – errors are given for the molecular mass measurement via mass photometry, but it is unclear what exactly the errors represent (standard deviation, standard error of the mean, etc). This should be clarified, and if multiple MP measurements were taken, n should be specified.
- Line 91, page 9 – this sentence is unclear. How is the ‘high correlation’ between the two nucleosome structures determined. Additionally, the sentence refers to crosslinked Melbournevirus NLPs, not the Medusavirus NLPs determined in the paper. Is this in error? If not, it should be made clear what Melbournevirus NLP structures are being referred to.
- Line 98, page 10 – the abbreviation ‘SHL’ should be defined.
- Line 125, page 11 – it is unclear how the conclusion that Medusavirus NLPs are structurally more like other viral NLPs is reached. This should be clarified in the text, particularly as it is key for the major message of the article.
- Figure S4 – please include a scale bar for 2D class averages.
- Please include the atomic B factors in the cryoEM table.

Reviewer #4

(Remarks to the Author)

Reviewer #5

(Remarks to the Author)

Version 1:

Reviewer comments:

Reviewer #1

(Remarks to the Author)

In their revised manuscript, the authors succeeded in improving the resolution of the cryo-EM map of the MM nucleosome from 4.3 Å to 3.3 Å, enabling a more detailed molecular model to be placed. By comparing the improved structure of the MM nucleosome with the previously reported MV nucleosome and eukaryotic nucleosome, the authors discuss the importance of nucleosomes in NCLDV. Regarding the main point of this article, namely the finding that MM's linker histone-like protein H1 (not found in other NCLDVs) does not function to bind nucleosomes like the usual H1, the revised version of the manuscript does not include analyses that might suggest the specific function(s) it possesses. I understand that such analyses are very difficult to undertake, yet I feel that even a hypothetical function for this unusual histone H1 would be an important element for this article.

Reviewer #2

(Remarks to the Author)

In the revised version, the authors have addressed some of our concerns, but the manuscript still lacks data on the role of these nucleosomes in viruses. While the structure is improved and more convincing, it is largely similar to known nucleosome structures. Some differences are present, which is expected due to the rapid evolution of viruses and the reduced selective pressure on histone sequences in the absence of the regulatory proteins that shape eukaryotic chromatin. Overall, the structural differences described are not striking enough to capture the attention of a general audience but will interest nucleosome structure specialists.

Without functional data, I believe this study is more suitable for a specialized journal rather than a broad-audience journal

like Nature Communications.

Reviewer #3

(Remarks to the Author)

All my major concerns have been addressed with this revision. In particular, the structural findings of the study are significantly strengthened with the newer high-resolution map. Several minor comments are given below.

Minor comments:

Please include the raw AUC data (with fits) used to generate Figure 2C as a supplementary figure.

There is discussion about a broad range of species that may be present in the AUC experiment (as highlighted by Reviewer 2). The molecular mass range of the MM-NLPs and e-Nucs determined by MP is similarly broad (Figure 2F). Is there likely to be a range of species present in these samples as well?

Table 1 - please note that 'S' in $s(20,w)$ should be lower case.

Line 286 - 'low sequence conservation compared to eukaryotic' - a little unclear, be specific which species, particularly as the sequence identity is given to three significant figures. Two species are listed.

Line 63 - 'decoupling of transcription from translation with the m7G capping' - 'the' is not necessary in this sentence. Alternatively, use 'cap' instead of 'capping'.

Line 121 - 'the only known viral nucleosome structure' suggest replacing 'known' with 'experimentally determined'.

Reviewer #4

(Remarks to the Author)

Reviewer #5

(Remarks to the Author)

Detailed response to all reviewers' comments:

Reviewer #1 (Remarks to the Author):

The manuscript “Characterization of Medusavirus encoded histones reveals nucleosome-like structures and a unique linker histone” (NCOMMS-24-22386) by Toner et al. reports the structure of the Medusavirus medusae (MM) nucleosome, which is known to encode the four core histones and linker histone H1 on separate genes. In this manuscript, the authors reconstituted the nucleosome-like structure of Medusavirus, a giant DNA virus, with its four histones and an artificial positioning sequence, determined the structure by cryo-EM at a resolution of 4.3 Å revealing a unique viral nucleosome-like structure. Nucleosome arrays were then reconstituted, and AFM was used to observe the formation of “beads-on-a-string” structures. In addition to the structural results, biochemical analyses using analytical ultracentrifugation and thermal stability assays, as well as reconstitution of the viral nucleosome using “random” sequences that are not strong positioning sequences, indicate that the nucleosome-like structures formed by viral histones are unstable compared with those of humans. This manuscript presents a pioneering study of viral nucleosome-like structures and their properties, which are still poorly understood. **However, the author’s assertions derived from cryo-EM results are poorly supported due to the low resolution of the cryo-structure.** Discussion of the interaction of amino acid side chains despite the low resolution seems somewhat hazardous. The data are also weak in terms of novelty and impact. Above all, **there is a lack of data on the role that these nucleosome-like structures might play in viruses.** Therefore, the manuscript may be unsuitable for publication in Nature Communications. The authors are strongly encouraged to respond to the following comments to improve the overall quality of the manuscript.

Resolution: We have now improved the overall resolution of our structure to 3.3 Å (see below).

Role of nucleosome in the virus: We fully agree that further research is necessary to determine the role of these nucleosome-like structures in Medusavirus. However, at this time we are unable to perform *in vivo* experiments as we do not have access to the virus (currently available in only one lab world-wide). We have shown previously that Melbournevirus histones localize to the cytoplasmic viral factory within the Amoeba host. Such experiments are currently

not possible in Medusavirus as the experimental system is not developed to that stage yet. For this reason, localization experiments similar to those we performed with Melbournevirus histones upon infection of Amoeba, and exploring potential viral genome regulation through Mass Spectrometry analysis of intact chromatin from virions is currently not possible.

Novelty: Previously, the only known viral nucleosome structure was that of Melbournevirus-NLP (7LV8 at 3.4 Å and 7N8N at 3.9 Å). Here, we present only the second viral nucleosome structure and highlight both similarity and differences in a comparison with Melbournevirus-NLP and eukaryotic nucleosomes. We believe that our data demonstrates the plasticity of histone sequences in its adaptation to unique virus requirements. The novelty lies in the comparison of these structures, and our work drives home the fact that the previously published structure is not just a curiosity but that giant viruses adapt histones to their purpose in various ways. The novelty also lies in our identification of a ‘linker histone like’ protein that appears to emphatically not function like a linker histone.

Major comments:

1. The resolution of the determined structure may not be high enough to assess the directions of the amino acid side chains discussed in Figures 4B and 4C, 5B, 5C and 5D. The authors should improve the quality of the maps to confirm their discussion. Alternatively, for example for Figures 4B and 5B, where the interaction between H4 and the C-tail of H3 is analyzed, the authors could perform a mutant assay to validate their assertion.

We put considerable effort into optimizing all aspects of our workflow and now present a map of MM-NLP_{207W} at 3.3 Å resolution (previous resolution 4.4 Å overall). The new map allowed for further refinement and better definition of the model, yet overall it remains very similar to the previous model (RMSD=1.086 Å). Importantly, with this increased resolution we are now able to resolve many side chains. As such, the text has been updated to highlight new insights. Specifically, pertaining to this reviewer comment, we are now able to resolve the helical structure of the H3 C-terminal. While the increased resolution didn’t change any of the overall conclusions, we appreciate this suggestion which allowed us to be less tentative in our conclusions. Figures 3, 4, and 5, S4, 5, and 6 are completely revised to reflect the new model and map, with density shown for key side chains (Figure 4b and 4c).

2. The authors compare the structure of the Medusavirus nucleosome with the Melbournevirus nucleosome previously determined and point out that the Medusavirus nucleosome is closer to the eukaryotic nucleosome. However, I don’t entirely agree with this affirmation. The structural differences are not obvious and, what’s more, are similar in several respects. The authors should focus on the functional differences, which could significantly strengthen their claim.

We agree that comparing how these distantly related viruses use histones throughout their life cycle would be a fascinating area of study, but, for reasons pointed out above, this is currently not possible as the challenges of each viral system are still being overcome. As NCLDV maintain a variety of different infection cycles that utilize the nucleus in different ways, each viral system must be considered to address experimental *in vivo* questions.

Our main point (which we believe we now articulate better) is that viral nucleosomes have common features that sets them apart from their eukaryotic counterparts. In addition, they also have distinct features that set them apart from each other. Of course, our ‘sample size’ is only two viral nucleosomes and, therefore, it is not yet possible to say what the common characteristics of a viral nucleosome are. To address this comment, we have revised this discussion throughout the manuscript.

3. Medusavirus medusae is a unique virus that possesses a binding histone H1, absent in other NCLDVs. The authors showed by AFM imaging that Medusavirus H1 lacks the ability to compact chromatin. As the authors show, Medusavirus H1 does not appear to be able to compact eukaryotic tri-nucleosomes. However, the function of Medusavirus H1 should be further analyzed by structural analyses involving Medusavirus tri-nucleosome. It would also be interesting to know whether acidic Medusavirus H1 could function as a viral chromatin protein; by performing EMSA or other methods, can purified Medusavirus H1 bind to viral nucleosome-like structures? the authors should show whether Medusavirus H1's

inability to bind to nucleosomes explains its inability to function as a compaction protein. From a general point of view, the authors should pursue the analysis of the Medusavirus H1-bound nucleosome in parallel with other biochemical experiments, and not only by discussing the cryo-EM structure they have determined, to fully understand the function of this viral nucleosome and its putative H1 histone.

In response to this very valid point, we performed additional assays to directly test the binding of Medusa H1 to DNA and viral nucleosome-like particles. Our data (now included in the manuscript) show that while Mouse H1.0 binds DNA and nucleosomes, MM H1 does not bind DNA or trinucleosomes. This data suggests that MM H1 does not compact nucleosome-like particles because it is not capable of the necessary interactions for compaction. This new data (shown here for reference) has been added to the manuscript (new Figure S7d and S7e)/

Minor comments:

1: P21, Sedimentation velocity analytical ultracentrifugation (SV-AUC):

In the text, it is cited that “To evaluate the homogeneity, molecular size, and molecular shape of MM-NLPs and 79 MM-LE-tri-NLP in solution; we used SV-AUC with absorbance optics ($\lambda = 280 \text{ nm}$). MM-NLP 80 of various DNA lengths and MM-tri-NLP, all at 250 nM, were spun at 30,000 – 35,000 rpm at 81 20°C in the Beckman XL-A ultracentrifuge” However, the analytical ultracentrifuge data in the manuscript appear to be based on 207W DNA. Please correct.

We thank the reviewer for catching this mistake. The methods section has now been updated to reflect that the AUC data represented NLP made with 207W DNA.

2: P24 L142-144

Please check that the model’s name and pixel size of the cryo-EM are correct.

Please also correct the defocus range “-2.2 μM ” to “ μm ”.

The typos in the equipment name and defocus have been corrected. Additionally, the pixel size has been updated for the new data that is now included after revision.

3: Please correct the following typos:

- P8 L48: “SYPRO Orange” and not “SYBRO Orange”.
- P24 L156: “ab initio model” and not “an initio model”.

We thank the reviewer for catching these typos, both as been corrected in the manuscript text.

Reviewer #2 (Remarks to the Author):

In this paper, the authors present the structure of the nucleosome from *Medusavirus medusae* (MM), which bears a resemblance to the canonical nucleosome but with certain distinct features. The resolution of the structure is somewhat limited, and the authors themselves acknowledge that some of the differences observed from the canonical nucleosome might be due to their inability to construct parts of the viral nucleosome. This limits the overall quality and reliability of the data. The study does not discuss biological function and significance of this viral nucleosome, which restricts the novelty of the research. In my view, the current state of the work may not meet the publication standards of Nature Communications.

See our response to reviewer 1

1. At one point, the authors mention an acidic patch, stating: “The negative charge in this region of the MM-NLP is less pronounced (Figure 4D), possibly due to our inability to construct parts of the H2A docking domain.” This raises concerns about the quality of the model and the potential impact on other regions. The authors should improve their structural analysis to obtain a higher resolution structure, enabling them to better build this and other parts of the nucleosome. This also pertains to Figures 4 and 5, where the authors discuss interactions between unique tails and loops in MM-NLP and interactions between the four-helix bundle (which is crucial for nucleosome stability) at a 4.3Å resolution structure, where side chains are not fully resolved. The authors need to demonstrate how the regions they discuss in the main figures fit into the experimental density. The authors should improve their structural analysis and the resolution of their maps to build a more reliable model.

We have improved the resolution of our structure to 3.3 Å overall. While many regions are now much better resolved (e.g. many side chains, the newly built H3 C-terminal α helix), the regions of the H2A docking domain that were previously ‘invisible’ are still not resolved. As we point out in the revised text, this is probably due to lacking interactions with the histone octamer surface (unlike the docking domain of Melbournevirus nucleosomes, where this region displays interactions that are not seen in eNuc.). We have now included a figure to highlight this point (Figure 5b). As such, our inability to observe this region of the docking domain in *Medusavirus* nucleosomes probably represents a ‘feature, not a bug’. As pointed out in our response to reviewer 1, we have remade all figures with the new model and map with density shown for key side chains (Figure 4b and 4c).

2. The authors demonstrate that the putative linker histone h1 does not compact tri-nucleosomes due to the absence of crucial compaction residues and an acidic electrostatic surface. It would be beneficial if the authors could demonstrate whether this histone-like protein binds to chromatin both *in vitro* and *in vivo*. From their experiments, it’s not even apparent if this linker histone-like protein can bind to the viral nucleosome *in vitro*.

We have performed additional DNA and nucleosome binding experiments that are now included in the revised manuscript (see also response to reviewer 1).

3. In Table 1, the experimental molecular weight for MM-NLP displays a large range (238-343). Considering the authors’ data indicates low stability of the octamer (Figure S3) and nucleosome (Figure S3, Figure 2), did the authors consider different possibilities for the observed experimental molecular weight of the sample? It’s possible that their sample could be a mixture of nucleosomes, hexasomes, or even tetrasomes.

This is an astute observation. They are correct that the reported molecular weight range of the sample suggests a mixture of nucleosomes, hexasome, or even tetrasomes. Upon performing the experiment, we did identify the largest population within the sample (91 %) as the MM-NLP, with the reported experimental molecular weight of 291 kDa, with minor populations of potential sub-nucleosomal species. For transparency, we report the molecular weight range for the entire sample along with the largest population within that range. We have updated the text to include this discussion.

4. The authors suggest the Viral Eukaryogenesis hypothesis in several parts of the text, which is purely speculative and the

data presented by the authors do not support this hypothesis any more than they do horizontal gene transfer. The data presented in this manuscript are insufficient to support any evolutionary theory.

We agree that the data presented in this manuscript are insufficient to support any evolutionary theory and, as such, we have attempted to simply include relevant theories regarding our organism of interest. In light of the reviewer's comment, we have revised our previous mention of both viral eukaryogenesis and horizontal gene transfer through the manuscript. We feel that mentioning the viral eukaryogenesis hypothesis provides important context for the manuscript, but we do not wish to imply that our data supports either one.

5. Could the authors elaborate on how this MM-NLP differs from other NCLDV NLPs?

Currently, the only other known viral particle is the Melbournevirus-NLP (MV-NLP). In the manuscript, we included comparisons between the MV-NLP and highlight the main differences and similarities. Based on this reviewer's comment, we have highlighted the limited ability of NCLDV NLPs data and we feel this emphasizes the novelty of the manuscript for the reader.

Minor points:

1. Would it be possible for the authors to display the complete MM-NLP structure in the same orientation as in Figure 4E? This would allow for a better understanding of the contribution of the acidic H3 C-tail.

Thank you for the suggestion. This additional figure panel is now included as Figure S5d.

2. The authors propose that the N-terminal tail of H4 in Melbournevirus performs the same function as the MM H3 C-tail. Does the Melbournevirus H4 N-tail exhibit a similar charged surface as depicted for the MM-H3 tail?

We have made this comparison, which is now included as Figure S5b. Indeed, the surfaces of the viral particles differ substantially due to these differences in structure.

3. The authors mention that *A. castellanii* possesses three linker histones. Is there any available data demonstrating that these proteins bind to chromatin *in vivo*?

Currently, there is no available data investigating Amoeba linker histones (and, further, amoeba chromatin) *in vitro* or *in vivo*. At this reviewer's suggestion, we completed binding analysis of Amoeba H1.1 to DNA and viral nucleosome-like particles. We decided to test the Amoeba H1.1, as it was similar in charge to the MM-H1 and similar length to eH1. The 163 AA length of Amoeba H1.1 is more closely related to eH1 (190 AA) than the other Amoeba H1 variants (332 AA and 514 AA). We tested the ability of Amoeba H1.1 to bind to DNA using Fluorescence Polarization (FP). Shown in the new Figure S7, and also below, together with the Mouse H1.0 and MM H1 data, this data demonstrates that the Amoeba H1.1 does not bind DNA. As such, it is possible that this particular amoeba H1 fulfils different functions (as we suspect for virus-encoded H1). Incidentally, alphafold predictions of the three amoeba H1 sequences all show a globular domain that is very similar to that of domain 1 of Medusavirus H1, and to eukaryotic linker histone. The three amoeba H1 histones differ widely in their additional sequence elements. This discussion was added in the manuscript text.

4. Could the authors provide a clearer explanation of the connection between the H3 helix and the tethering of the H2AH2B dimer to the h3H4 tetramer? The current explanation is somewhat confusing.

In the new higher-resolution structure where side chains are clearly resolved, we do not see direct interactions between these regions, even though the main chains are in close proximity. We have revised the text as direct interactions are not supported by the model. If the reviewer is talking about the H2A-H2B docking domain and its interactions with the (H3-H4)₂, we have included a new figure in the revised version detailing these interactions

5. Would it be possible for the authors to indicate the location of the H3-H3' and H2B-H4 four-helix bundle on the model of the nucleosome structure in Figure 5, to help the reader's orientation (as they did in Figure 4)?

To provide information on orientation, we added labels indicating the location of both the H3-H3' and H2B-H4 four helix bundles in the disk-view (face-view) of the MM-NLP in Figure 3b.

Reviewer #3 (Remarks to the Author):

This article from Toner et al presents the structure of nucleosome-like proteins from the virus Medusavirus medusase, highlighting differences between the Medusavirus nucleosomes, eukaryotic nucleosomes and nucleosomes from the Melbournevirus genus of viruses. This article is generally well-written, and the conclusions of the article are supported by the data presented. It is unclear how one of the major conclusions of the paper, that Medusavirus NLPs are structurally more similar to eukaryotic nucleosomes, is reached, so this could be better clarified in the article, particularly as it appears to be key to the main message of the article. The results overall provide new structural insights into nucleosome/histone evolution, and the article is suitable for publication in Nature Communications once the below suggested changes are addressed.

My major concerns are:

- Please deposit the lower-resolution un-fixed map in the EMDB and include the deposition code in the manuscript, as this map was discussed extensively in the manuscript.

We have deposited the lower-resolution un-fixed Medusavirus nucleosome data into EMDB (EMD-4598), alongside the new 3.3 Å map for crosslinked MM-NLP. The PDB entry for the 3.3 Å model has also been updated.

- Line 62, page 6 – errors are given for the molecular mass measurement via mass photometry, but it is unclear what exactly the errors represent (standard deviation, standard error of them mean, etc). This should be clarified, and if multiple MP measurements were taken, n should be specified.

Error for measurements for mass photometry refer to standard deviation of the Gaussian function fit for the main species corresponding to the respective mass at the center of each peak. This information is included in the figure legend (Figure 2f). Additionally, we have clarified in the figure legend that this is one representative data set of triplicate measurements.

- Line 91, page 9 – this sentence is unclear. How is the ‘high correlation’ between the two nucleosome structures determined. Additionally, the sentence refers to crosslinked Melbournevirus NLPs, not the Medusavirus NLPs determined in the paper. Is this in error? If not, it should be made clear what Melbournevirus NLP structures are being referred to.

The correlation between the two maps was determined using the Chimera “Fit in map” feature. To this point, the main text has been clarified and how we determined this value was added to the methods.

The reviewer is correct that this sentence is referring only to Medusavirus and this has been corrected.

- Line 98, page 10 – the abbreviation ‘SHL’ should be defined.

The abbreviation is now clearly defined as Super Helical Location (SHL).

- Line 125, page 11 – it is unclear how the conclusion that Medusavirus NLPs are structurally more like other viral NLPs is reached. This should be clarified in the text, particularly as it is key for the major message of the article.

See our response to reviewer 1. Major revisions were made in the text.

- Figure S4 – please include a scale bar for 2D class averages.

We have included a scale bar for 2D class averages for both datasets.

- Please include the atomic B factors in the cryoEM table.

We have included the atomic B factors in the cryoEM table.

Reviewer #4 (Remarks to the Author):

We thank Reviewer #4 for their contribution to the peer review of our manuscript. We hope that we addressed all of your concerns in the revised manuscript.

Reviewer #5 (Remarks to the Author):

We thank Reviewer #5 for their contribution to peer review of our manuscript. We hope that we addressed all of your concerns in the revised manuscript.

Dear Professor Luger,

Thank you for submitting your manuscript "Characterization of Medusavirus encoded histones reveals nucleosome-like structures and a unique linker histone" to Nature Communications. I am delighted to say that we are happy, in principle, to publish it under an open access license.

First, we ask you to revise your paper to address our editorial requests (in the attached Author Checklist) and any remaining comments from reviewers (included at the end of this email, if applicable). Please note that we have decided to overrule the concerns on novelty and the request for data on the role of these nucleosomes in medusaviruses.

REVIEWERS' COMMENTS

Reviewer #1 (Remarks to the Author):

In their revised manuscript, the authors succeeded in improving the resolution of the cryo-EM map of the MM nucleosome from 4.3 Å to 3.3 Å, enabling a more detailed molecular model to be placed. By comparing the improved structure of the MM nucleosome with the previously reported MV nucleosome and eukaryotic nucleosome, the authors discuss the importance of nucleosomes in NCLDV. Regarding the main point of this article, namely the finding that MM's linker histone-like protein H1 (not found in other NCLDVs) does not function to bind nucleosomes like the usual H1, the revised version of the manuscript does not include analyses that might suggest the specific function(s) it possesses. I understand that such analyses are very difficult to undertake, yet I feel that even a hypothetical function for this unusual histone H1 would be an important element for this article.

We thank Reviewer #1 for this suggestion. We have added discussion of the potential function of MM-H1, particularly highlighting its early expression in the virus life cycle and proposed ideas in the field.

Reviewer #2 (Remarks to the Author):

In the revised version, the authors have addressed some of our concerns, but the manuscript still lacks data on the role of these nucleosomes in viruses. While the structure is improved and more convincing, it is largely similar to known nucleosome structures. Some differences are present, which is expected due to the rapid evolution of viruses and the reduced selective pressure on histone sequences in the absence of the regulatory proteins that shape eukaryotic chromatin. Overall, the structural differences described are not striking enough to capture the attention of a general audience but will interest nucleosome structure specialists. Without functional data, I believe this study is more suitable for a specialized journal rather than a broad-audience journal like Nature Communications.

We thank Reviewer #2 for their contribution to the peer review of our manuscript. While we are unable to complete these experiments at this time, we hope that technological advances will allow us to grow and manipulate the virus in the future.

Reviewer #3 (Remarks to the Author):

All my major concerns have been addressed with this revision. In particular, the structural findings of the study are significantly strengthened with the newer high-resolution map. Several minor comments are given below.

Minor comments:

- 1) Please include the raw AUC data (with fits) used to generate Figure 2C as a supplementary figure.

For full data availability and transparency, we have included the raw AUC data (with fits) used to generate Figure 2C and Table 1 in the source data file.

- 2) There is discussion about a broad range of species that may be present in the AUC experiment (as highlighted by Reviewer 2). The molecular mass range of the MM-NLPs and e-Nucs determined by MP is similarly broad (Figure 2F). Is there likely to be a range of species present in these samples as well?

While a range of molecular weights are represented in the Gaussian fit, we cannot differentiate what is due to noise and potential sub-species. In mass photometry, larger molecules exhibit broader peaks due to weaker contrast signals, which introduce more noise and reduce resolution. Thus, the observed standard deviations are expected for molecules of this size. However, any sub-species present would represent a low percentage of the sample since they do not form separate peaks or skew the Gaussian fit. Notably, the MP data is of tri-nucleosomes and show no mono- or di-nucleosome sub-species, which can be resolved as distinct peaks.

- 3) Table 1 - please note that 'S' in s(20,w) should be lower case.

This typo has been corrected.

- 4) Line 286 - 'low sequence conservation compared to eukaryotic' - a little unclear, be specific which species, particularly as the sequence identity is given to three significant figures. Two species are listed.

This sentence has been clarified that the comparison is to *X. laevis* H1.

- 5) Line 63 - 'decoupling of transcription from translation with the m7G capping' - 'the' is not necessary in this sentence. Alternatively, use 'cap' instead of 'capping'.

This typo has been corrected.

- 6) Line 121 - 'the only known viral nucleosome structure' suggest replacing 'known' with 'experimentally determined'.

This change has been implemented in the text.

Reviewer #4 (Remarks to the Author):

We thank Reviewer #4 for their contribution to the peer review of our manuscript. We hope that we addressed all of your concerns in the revised manuscript.

Reviewer #5 (Remarks to the Author):

We thank Reviewer #5 for their contribution to the peer review of our manuscript. We hope that we addressed all of your concerns in the revised manuscript.